# A Time Series is Worth 64 Words:
# Long-term Forecasting with Transformers

**Yuqi Nie**[1]*, **Nam H. Nguyen**[2]*, **Phanwadee Sinthong**[2], **Jayant Kalagnanam**[2]
[1]Princeton University [2]IBM Research
`ynie@princeton.edu, nnguyen@us.ibm.com, Gift.Sinthong@ibm.com,`
`jayant@us.ibm.com`

## Abstract

We propose an efficient design of Transformer-based models for multivariate time series forecasting and self-supervised representation learning. It is based on two key components: (i) segmentation of time series into subseries-level patches which are served as input tokens to Transformer; (ii) channel-independence where each channel contains a single univariate time series that shares the same embedding and Transformer weights across all the series. Patching design naturally has three-fold benefit: local semantic information is retained in the embedding; computation and memory usage of the attention maps are quadratically reduced given the same look-back window; and the model can attend longer history. Our channel-independent patch time series Transformer (PatchTST) can improve the long-term forecasting accuracy significantly when compared with that of SOTA Transformer-based models. We also apply our model to self-supervised pre-training tasks and attain excellent fine-tuning performance, which outperforms supervised training on large datasets. Transferring of masked pre-trained representation on one dataset to others also produces SOTA forecasting accuracy.

## 1 Introduction

Forecasting is one of the most important tasks in time series analysis. With the rapid growth of deep learning models, the number of research works has increased significantly on this topic (Bryan & Stefan, 2021; Torres et al., 2021; Lara-Benítez et al., 2021). Deep models have shown excellent performance not only on forecasting tasks, but also on representation learning where abstract representation can be extracted and transferred to various downstream tasks such as classification and anomaly detection to attain state-of-the-art performance.

Among deep learning models, Transformer has achieved great success on various application fields such as natural language processing (NLP) (Kalyan et al., 2021), computer vision (CV) (Khan et al., 2021), speech (Karita et al., 2019), and more recently time series (Wen et al., 2022), benefiting from its attention mechanism which can automatically learn the connections between elements in a sequence, thus becomes ideal for sequential modeling tasks. Informer (Zhou et al., 2021), Autoformer (Wu et al., 2021), and FEDformer (Zhou et al., 2022) are among the best variants of the Transformer model successfully applying to time series data. Unfortunately, regardless of the complicated design of Transformer-based models, it is shown in the recent paper (Zeng et al., 2022) that a very simple linear model can outperform all of the previous models on a variety of common benchmarks and it challenges the usefulness of Transformer for time series forecasting. In this paper, we attempt to answer this question by proposing a channel-independence patch time series Transformer (PatchTST) model that contains two key designs:

- **Patching.** Time series forecasting aims to understand the correlation between data in each different time steps. However, a single time step does not have semantic meaning like a word in a sentence, thus extracting local semantic information is essential in analyzing their connections. Most of the previous works only use point-wise input tokens, or just a handcrafted information

---

*Equal contribution.

| Models | $L$ | $N$ | patch | method | MSE |
|---|---|---|---|---|---|
| Channel-independent PatchTST | 96 | 96 | | | 0.518 |
| | 380 | 96 | | down-sampled | 0.447 |
| | 336 | 336 | | | 0.397 |
| | 336 | 42 | ✓ | | 0.367 |
| | 336 | 42 | ✓ | self-supervised | **0.349** |
| Channel-mixing FEDFormer | 336 | 336 | | | 0.597 |
| DLinear | 336 | 336 | | | 0.410 |

| Running time (s) with $L = 336$ | | | |
|---|---|---|---|
| Dataset | w. patch | w.o. patch | Gain |
| Traffic | 464 | 10040 | x 22 |
| Electricity | 300 | 5730 | x 19 |
| Weather | 156 | 680 | x 4 |

Table 1: A case study of multivariate time series forecasting on Traffic dataset. The prediction horizon is 96. Results with different look-back window $L$ and number of input tokens $N$ are reported. The best result is in **bold** and the second best is underlined. Down-sampled means sampling every 4 step and adding the last value. All the results are from supervised training except the best result which uses self-supervised learning.

from series. In contrast, we enhance the locality and capture comprehensive semantic information that is not available in point-level by aggregating time steps into subseries-level patches.

- **Channel-independence.** A multivariate time series is a multi-channel signal, and each Transformer input token can be represented by data from either a single channel or multiple channels. Depending on the design of input tokens, different variants of the Transformer architecture have been proposed. Channel-mixing refers to the latter case where the input token takes the vector of all time series features and projects it to the embedding space to mix information. On the other hand, channel-independence means that each input token only contains information from a single channel. This was proven to work well with CNN (Zheng et al., 2014) and linear models (Zeng et al., 2022), but hasn't been applied to Transformer-based models yet.

We offer a snapshot of our key results in Table 1 by doing a case study on Traffic dataset, which consists of 862 time series. Our model has several advantages:

1. Reduction on time and space complexity: The original Transformer has $O(N^2)$ complexity on both time and space, where $N$ is the number of input tokens. Without pre-processing, $N$ will have the same value as input sequence length $L$, which becomes a primary bottleneck of computation time and memory in practice. By applying patching, we can reduce $N$ by a factor of the stride: $N \approx L/S$, thus reducing the complexity quadratically. Table 1 illustrates the usefulness of patching. By setting patch length $P = 16$ and stride $S = 8$ with $L = 336$, the training time is significantly reduced as much as 22 time on large datasets.

2. Capability of learning from longer look-back window: Table 1 shows that by increasing look-back window $L$ from 96 to 336, MSE can be reduced from 0.518 to 0.397. However, simply extending $L$ comes at the cost of larger memory and computational usage. Since time series often carries heavily temporal redundant information, some previous work tried to ignore parts of data points by using downsampling or a carefully designed sparse connection of attention (Li et al., 2019) and the model still yields sufficient information to forecast well. We study the case when $L = 380$ and the time series is sampled every 4 steps with the last point added to sequence, resulting in the number of input tokens being $N = 96$. The model achieves better MSE score (0.447) than using the data sequence containing the most recent 96 time steps (0.518), indicating that longer look-back window conveys more important information even with the same number of input tokens. This leads us to think of a question: is there a way to avoid throwing values while maintaining a long look-back window? Patching is a good answer to it. It can group local time steps that may contain similar values while at the same time enables the model to reduce the input token length for computational benefit. As evident in Table 1, MSE score is further reduced from 0.397 to 0.367 with patching when $L = 336$.

3. Capability of representation learning: With the emergence of powerful self-supervised learning techniques, sophisticated models with multiple non-linear layers of abstraction are required to capture abstract representation of the data. Simple models like linear ones (Zeng et al., 2022) may not be preferred for that task due to its limited expressibility. With our PatchTST model, we not only confirm that Transformer is actually effective for time series forecasting, but also demonstrate the representation capability that can further enhance the forecasting performance. Our PatchTST has achieved the best MSE (0.349) in Table 1.

We introduce our approach in more detail and conduct extensive experiments in the following sections to conclusively prove our claims. We not only demonstrate the model effectiveness with supervised forecasting results and ablation studies, but also achieves SOTA self-supervised representation learning and transfer learning performance.

## 2 RELATED WORK

**Patch in Transformer-based Models.** Transformer (Vaswani et al., 2017) has demonstrated a significant potential on different data modalities. Among all applications, patching is an essential part when local semantic information is important. In NLP, BERT (Devlin et al., 2018) considers subword-based tokenization (Schuster & Nakajima, 2012) instead of performing character-based tokenization. In CV, Vision Transformer (ViT) (Dosovitskiy et al., 2021) is a milestone work that splits an image into $16 \times 16$ patches before feeding into the Transformer model. The following influential works such as BEiT (Bao et al., 2022) and masked autoencoders (He et al., 2021) are all using patches as input. Similarly, in speech researchers are using convolutions to extract information in sub-sequence levels from raw audio input (Baevski et al., 2020; Hsu et al., 2021).

**Transformer-based Long-term Time Series Forecasting.** There is a large body of work that tries to apply Transformer models to forecast long-term time series in recent years. We here summarize some of them. LogTrans (Li et al., 2019) uses convolutional self-attention layers with LogSparse design to capture local information and reduce the space complexity. Informer (Zhou et al., 2021) proposes a ProbSparse self-attention with distilling techniques to extract the most important keys efficiently. Autoformer (Wu et al., 2021) borrows the ideas of decomposition and auto-correlation from traditional time series analysis methods. FEDformer (Zhou et al., 2022) uses Fourier enhanced structure to get a linear complexity. Pyraformer (Liu et al., 2022) applies pyramidal attention module with inter-scale and intra-scale connections which also get a linear complexity.

Most of these models focus on designing novel mechanisms to reduce the complexity of original attention mechanism, thus achieving better performance on forecasting, especially when the prediction length is long. However, most of the models use point-wise attention, which ignores the importance of patches. LogTrans (Li et al., 2019) avoids a point-wise dot product between the key and query, but its value is still based on a single time step. Autoformer (Wu et al., 2021) uses autocorrelation to get patch level connections, but it is a handcrafted design which doesn't include all the semantic information within a patch. Triformer (Cirstea et al., 2022) proposes patch attention, but the purpose is to reduce complexity by using a pseudo timestamp as the query within a patch, thus it neither treats a patch as a input unit, nor reveals the semantic importance behind it.

**Time Series Representation Learning.** Besides supervised learning, self-supervised learning is also an important research topic since it has shown the potential to learn useful representations for downstream tasks. There are many non-Transformer-based models proposed in recent years to learn representations in time series (Franceschi et al., 2019; Tonekaboni et al., 2021; Yang & Hong, 2022; Yue et al., 2022). Meanwhile, Transformer is known to be an ideal candidate towards foundation models (Bommasani et al., 2021) and learning universal representations. However, although people have made attempts on Transformer-based models like time series Transformer (TST) (Zerveas et al., 2021) and TS-TCC (Eldele et al., 2021), the potential is still not fully realized yet.

## 3 PROPOSED METHOD

### 3.1 MODEL STRUCTURE

We consider the following problem: given a collection of multivariate time series samples with lookback window $L : (\boldsymbol{x}_1, ..., \boldsymbol{x}_L)$ where each $\boldsymbol{x}_t$ at time step $t$ is a vector of dimension $M$, we would like to forecast $T$ future values $(\boldsymbol{x}_{L+1}, ..., \boldsymbol{x}_{L+T})$. Our PatchTST is illustrated in Figure 1 where the model makes use of the vanilla Transformer encoder as its core architecture.

**Forward Process.** We denote a $i$-th univariate series of length $L$ starting at time index 1 as $\boldsymbol{x}_{1:L}^{(i)} = (x_1^{(i)}, ..., x_L^{(i)})$ where $i = 1, ..., M$. The input $(\boldsymbol{x}_1, ..., \boldsymbol{x}_L)$ is split to $M$ univariate series $\boldsymbol{x}^{(i)} \in \mathbb{R}^{1 \times L}$, where each of them is fed independently into the Transformer backbone according to

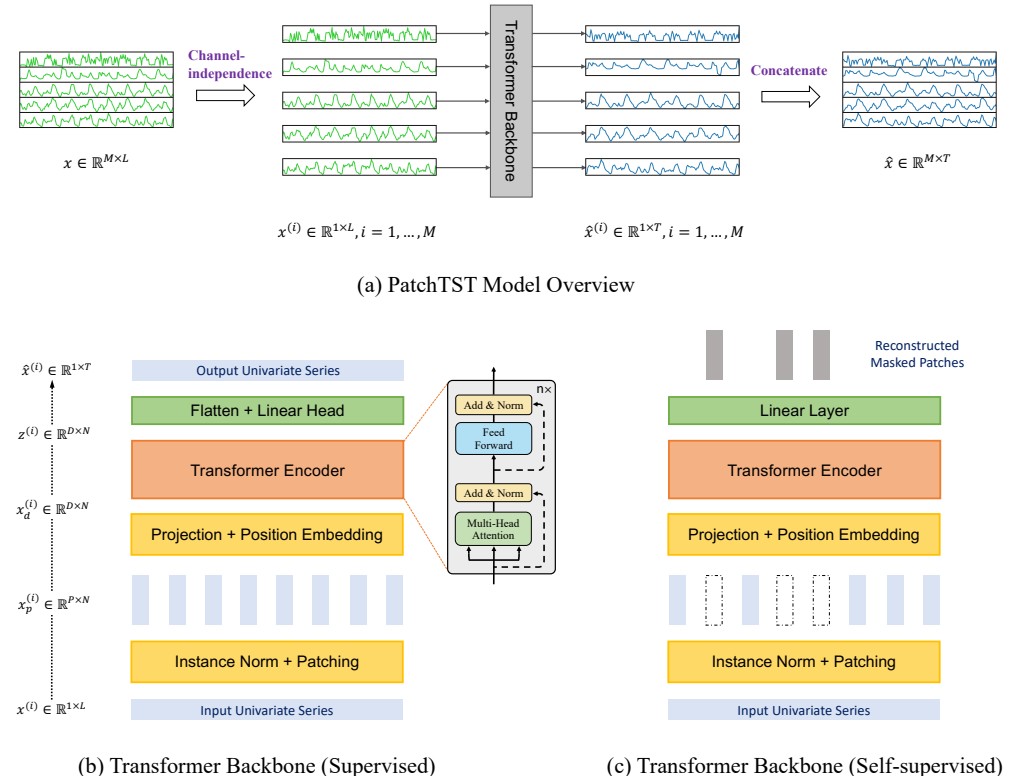

Figure 1: PatchTST architecture. (a) Multivariate time series data is divided into different channels. They share the same Transformer backbone, but the forward processes are independent. (b) Each channel univariate series is passed through instance normalization operator and segmented into patches. These patches are used as Transformer input tokens. (c) Masked self-supervised representation learning with PatchTST where patches are randomly selected and set to zero. The model will reconstruct the masked patches.

our channel-independence setting. Then the Transformer backbone will provide prediction results $\hat{\boldsymbol{x}}^{(i)} = (\hat{x}_{L+1}^{(i)}, ..., \hat{x}_{L+T}^{(i)}) \in \mathbb{R}^{1 \times T}$ accordingly .

**Patching.** Each input univariate time series $\boldsymbol{x}^{(i)}$ is first divided into patches which can be either overlapped or non-overlapped. Denote the patch length as $P$ and the stride - the non overlapping region between two consecutive patches as $S$, then the patching process will generate the a sequence of patches $\boldsymbol{x}_p^{(i)} \in \mathbb{R}^{P \times N}$ where $N$ is the number of patches, $N = \lfloor \frac{(L-P)}{S} \rfloor + 2$. Here, we pad $S$ repeated numbers of the last value $x_L^{(i)} \in \mathbb{R}$ to the end of the original sequence before patching.

With the use of patches, the number of input tokens can reduce from $L$ to approximately $L/S$. This implies the memory usage and computational complexity of the attention map are quadratically decreased by a factor of $S$. Thus constrained on the training time and GPU memory, patch design can allow the model to see the longer historical sequence, which can significantly improve the forecasting performance, as demonstrated in Table 1.

**Transformer Encoder.** We use a vanilla Transformer encoder that maps the observed signals to the latent representations. The patches are mapped to the Transformer latent space of dimension $D$ via a trainable linear projection $\mathbf{W}_p \in \mathbb{R}^{D \times P}$, and a learnable additive position encoding $\mathbf{W}_{\text{pos}} \in \mathbb{R}^{D \times N}$ is applied to monitor the temporal order of patches: $\boldsymbol{x}_d^{(i)} = \mathbf{W}_p \boldsymbol{x}_p^{(i)} + \mathbf{W}_{\text{pos}}$, where $\boldsymbol{x}_d^{(i)} \in \mathbb{R}^{D \times N}$ denote the input that will be fed into Transformer encoder in Figure 1. Then each head $h = 1, ..., H$ in multi-head attention will transform them into query matrices $Q_h^{(i)} = (\boldsymbol{x}_d^{(i)})^T \mathbf{W}_h^Q$, key matrices $K_h^{(i)} = (\boldsymbol{x}_d^{(i)})^T \mathbf{W}_h^K$ and value matrices $V_h^{(i)} = (\boldsymbol{x}_d^{(i)})^T \mathbf{W}_h^V$, where $\mathbf{W}_h^Q, \mathbf{W}_h^K \in \mathbb{R}^{D \times d_k}$ and

$\mathbf{W}_h^V \in \mathbb{R}^{D \times D}$. After that a scaled production is used for getting attention output $\mathbf{O}_h^{(i)} \in \mathbb{R}^{D \times N}$:

$$(\mathbf{O}_h^{(i)})^T = \text{Attention}(Q_h^{(i)}, K_h^{(i)}, V_h^{(i)}) = \text{Softmax}(\frac{Q_h^{(i)} K_h^{(i)T}}{\sqrt{d_k}}) V_h^{(i)}$$

The multi-head attention block also includes BatchNorm [1] layers and a feed forward network with residual connections as shown in Figure 1. Afterwards it generates the representation denoted as $\boldsymbol{z}^{(i)} \in \mathbb{R}^{D \times N}$. Finally a flatten layer with linear head is used to obtain the prediction result $\hat{\boldsymbol{x}}^{(i)} = (\hat{x}_{L+1}^{(i)}, ..., \hat{x}_{L+T}^{(i)}) \in \mathbb{R}^{1 \times T}$.

**Loss Function.** We choose to use the MSE loss to measure the discrepancy between the prediction and the ground truth. The loss in each channel is gathered and averaged over $M$ time series to get the overall objective loss: $\mathcal{L} = \mathbb{E}_{\boldsymbol{x}} \frac{1}{M} \sum_{i=1}^{M} \|\hat{\boldsymbol{x}}_{L+1:L+T}^{(i)} - \boldsymbol{x}_{L+1:L+T}^{(i)}\|_2^2$.

**Instance Normalization.** This technique has recently been proposed to help mitigating the distribution shift effect between the training and testing data (Ulyanov et al., 2016; Kim et al., 2022). It simply normalizes each time series instance $\boldsymbol{x}^{(i)}$ with zero mean and unit standard deviation. In essence, we normalize each $\boldsymbol{x}^{(i)}$ before patching and the mean and deviation are added back to the output prediction.

## 3.2 REPRESENTATION LEARNING

Self-supervised representation learning has become a popular approach to extract high level abstract representation from unlabelled data. In this section, we apply PatchTST to obtain useful representation of the multivariate time series. We will show that the learnt representation can be effectively transferred to forecasting tasks. Among popular methods to learn representation via self-supervise pre-training, masked autoencoder has been applied successfully to NLP (Devlin et al., 2018) and CV (He et al., 2021) domains. This technique is conceptually simple: a portion of input sequence is intentionally removed at random and the model is trained to recover the missing contents.

Masked encoder has been recently employed in time series and delivered notable performance on classification and regression tasks (Zerveas et al., 2021). The authors proposed to apply the multivariate time series to Transformer, where each input token is a vector $\boldsymbol{x}_i$ consisting of time series values at time step $i$-th. Masking is placed randomly within each time series and across different series. However, there are two potential issues with this setting: First, masking is applied at the level of single time steps. The masked values at the current time step can be easily inferred by interpolating with the immediate proceeding or succeeding time values without high level understanding of the entire sequence, which deviates from our goal of learning important abstract representation of the whole signal. Zerveas et al. (2021) proposed complex randomization strategies to resolve the problem in which groups of time series with different sizes are randomly masked.

Second, the design of the output layer for forecasting task can be troublesome. Given the representation vectors $\boldsymbol{z}_t \in \mathbb{R}^D$ corresponding to all $L$ time steps, mapping these vectors to the output containing $M$ variables each with prediction horizon $T$ via a linear map requires a parameter matrix $\mathbf{W}$ of dimension $(L \cdot D) \times (M \cdot T)$. This matrix can be particularly oversized if either one or all of these four values are large. This may cause overfitting when the number of downstream training samples is scarce.

Our proposed PatchTST can naturally overcome the aforementioned issues. As shown in Figure 1, we use the same Transformer encoder as the supervised settings. The prediction head is removed and a $D \times P$ linear layer is attached. As opposed to supervised model where patches can be overlapped, we divide each input sequence into regular non-overlapping patches. It is for convenience to ensure observed patches do not contain information of the masked ones. We then select a subset of the patch indices uniformly at random and mask the patches according to these selected indices with zero values. The model is trained with MSE loss to reconstruct the masked patches.

We emphasize that each time series will have its own latent representation that are cross-learned via a shared weight mechanism. This design can allow the pre-training data to contain different number of time series than the downstream data, which may not be feasible by other approaches.

---

[1] Zerveas et al. (2021) has shown that BatchNorm outperforms LayerNorm in time series Transformer.

## 4 EXPERIMENTS

### 4.1 LONG-TERM TIME SERIES FORECASTING

**Datasets.** We evaluate the performance of our proposed PatchTST on 8 popular datasets, including Weather, Traffic, Electricity, ILI and 4 ETT datasets (ETTh1, ETTh2, ETTm1, ETTm2). These datasets have been extensively utilized for benchmarking and publicly available on (Wu et al., 2021). The statistics of those datasets are summarized in Table 2. We would like to highlight several large datasets: Weather, Traffic, and Electricity. They have many more number of time series, thus the results would be more stable and less susceptible to overfitting than other smaller datasets.

| Datasets | Weather | Traffic | Electricity | ILI | ETTh1 | ETTh2 | ETTm1 | ETTm2 |
|---|---|---|---|---|---|---|---|---|
| Features | 21 | 862 | 321 | 7 | 7 | 7 | 7 | 7 |
| Timesteps | 52696 | 17544 | 26304 | 966 | 17420 | 17420 | 69680 | 69680 |

Table 2: Statistics of popular datasets for benchmark.

**Baselines and Experimental Settings.** We choose the SOTA Transformer-based models, including FEDformer (Zhou et al., 2022), Autoformer (Wu et al., 2021), Informer (Zhou et al., 2021), Pyraformer (Liu et al., 2022), LogTrans (Li et al., 2019), and a recent non-Transformer-based model DLinear (Zeng et al., 2022) as our baselines. All of the models are following the same experimental setup with prediction length $T \in \{24, 36, 48, 60\}$ for ILI dataset and $T \in \{96, 192, 336, 720\}$ for other datasets as in the original papers. We collect baseline results from Zeng et al. (2022) with the default look-back window $L = 96$ for Transformer-based models, and $L = 336$ for DLinear. But in order to avoid under-estimating the baselines, we also run FEDformer, Autoformer and Informer for six different look-back window $L \in \{24, 48, 96, 192, 336, 720\}$, and always choose the best results to create strong baselines. More details about the baselines could be found in Appendix A.1.2. We calculate the MSE and MAE of multivariate time series forecasting as metrics.

**Model Variants.** We propose two versions of PatchTST in Table 3. PatchTST/64 implies the number of input patches is 64, which uses the look-back window $L = 512$. PatchTST/42 means the number of input patches is 42, which has the default look-back window $L = 336$. Both of them use patch length $P = 16$ and stride $S = 8$. Thus, we could use PatchTST/42 as a fair comparison to DLinear and other Transformer-based models, and PatchTST/64 to explore even better results on larger datasets. More experimental details are provided in Appendix A.1.

**Results.** Table 3 shows the multivariate long-term forecasting results. Overall, our model outperform all baseline methods. Quantitatively, compared with the best results that Transformer-based models can offer, PatchTST/64 achieves an overall **21.0**% reduction on MSE and **16.7**% reduction on MAE, while PatchTST/42 attains an overall **20.2**% reduction on MSE and **16.4**% reduction on MAE. Compared with the DLinear model, PatchTST can still outperform it in general, especially on large datasets (Weather, Traffic, Electricity) and ILI dataset. We also experiment with univariate datasets where the results are provided in Appendix A.3.

### 4.2 REPRESENTATION LEARNING

In this section, we conduct experiments with masked self-supervised learning where we set the patches to be non-overlapped. Otherwise stated, across all representation learning experiments the input sequence length is chosen to be 512 and patch size is set to 12, which results in 42 patches. We consider high masking ratio where 40% of the patches are masked with zero values. We first apply self-supervised pre-training on the datasets mentioned in Section 4.1 for 100 epochs. Once the pre-trained model on each dataset is available, we perform supervised training to evaluate the learned representation with two options: (a) linear probing and (b) end-to-end fine-tuning. With (a), we only train the model head for 20 epochs while freezing the rest of the network; With (b), we apply linear probing for 10 epochs to update the model head and then end-to-end fine-tuning the entire network for 20 epochs. It was proven that a two-step strategy with linear probing followed by fine-tuning can outperform only doing fine-tuning directly (Kumar et al., 2022). We select a few representative results on below, and a full benchmark can be found in Appendix A.5.

| Models | | PatchTST/64 | | PatchTST/42 | | DLinear | | FEDformer | | Autoformer | | Informer | | Pyraformer | | LogTrans | |
|---|---|---|---|---|---|---|---|---|---|---|---|---|---|---|---|---|---|
| Metric | | MSE | MAE | MSE | MAE | MSE | MAE | MSE | MAE | MSE | MAE | MSE | MAE | MSE | MAE | MSE | MAE |
| Weather | 96 | **0.149** | **0.198** | 0.152 | 0.199 | 0.176 | 0.237 | 0.238 | 0.314 | 0.249 | 0.329 | 0.354 | 0.405 | 0.896 | 0.556 | 0.458 | 0.490 |
| | 192 | **0.194** | **0.241** | 0.197 | 0.243 | 0.220 | 0.282 | 0.275 | 0.329 | 0.325 | 0.370 | 0.419 | 0.434 | 0.622 | 0.624 | 0.658 | 0.589 |
| | 336 | **0.245** | **0.282** | 0.249 | 0.283 | 0.265 | 0.319 | 0.339 | 0.377 | 0.351 | 0.391 | 0.583 | 0.543 | 0.739 | 0.753 | 0.797 | 0.652 |
| | 720 | **0.314** | **0.334** | 0.320 | 0.335 | 0.323 | 0.362 | 0.389 | 0.409 | 0.415 | 0.426 | 0.916 | 0.705 | 1.004 | 0.934 | 0.869 | 0.675 |
| Traffic | 96 | **0.360** | **0.249** | 0.367 | 0.251 | 0.410 | 0.282 | 0.576 | 0.359 | 0.597 | 0.371 | 0.733 | 0.410 | 2.085 | 0.468 | 0.684 | 0.384 |
| | 192 | **0.379** | **0.256** | 0.385 | 0.259 | 0.423 | 0.287 | 0.610 | 0.380 | 0.607 | 0.382 | 0.777 | 0.435 | 0.867 | 0.467 | 0.685 | 0.390 |
| | 336 | **0.392** | **0.264** | 0.398 | 0.265 | 0.436 | 0.296 | 0.608 | 0.375 | 0.623 | 0.387 | 0.776 | 0.434 | 0.869 | 0.469 | 0.734 | 0.408 |
| | 720 | **0.432** | **0.286** | 0.434 | 0.287 | 0.466 | 0.315 | 0.621 | 0.375 | 0.639 | 0.395 | 0.827 | 0.466 | 0.881 | 0.473 | 0.717 | 0.396 |
| Electricity | 96 | **0.129** | **0.222** | 0.130 | 0.222 | 0.140 | 0.237 | 0.186 | 0.302 | 0.196 | 0.313 | 0.304 | 0.393 | 0.386 | 0.449 | 0.258 | 0.357 |
| | 192 | **0.147** | **0.240** | 0.148 | 0.240 | 0.153 | 0.249 | 0.197 | 0.311 | 0.211 | 0.324 | 0.327 | 0.417 | 0.386 | 0.443 | 0.266 | 0.368 |
| | 336 | **0.163** | **0.259** | 0.167 | 0.261 | 0.169 | 0.267 | 0.213 | 0.328 | 0.214 | 0.327 | 0.333 | 0.422 | 0.378 | 0.443 | 0.280 | 0.380 |
| | 720 | **0.197** | **0.290** | 0.202 | 0.291 | 0.203 | 0.301 | 0.233 | 0.344 | 0.236 | 0.342 | 0.351 | 0.427 | 0.376 | 0.445 | 0.283 | 0.376 |
| ILI | 24 | **1.319** | **0.754** | 1.522 | 0.814 | 2.215 | 1.081 | 2.624 | 1.095 | 2.906 | 1.182 | 4.657 | 1.449 | 1.420 | 2.012 | 4.480 | 1.444 |
| | 36 | 1.579 | 0.870 | **1.430** | **0.834** | 1.963 | 0.963 | 2.516 | 1.021 | 2.585 | 1.038 | 4.650 | 1.463 | 7.394 | 2.031 | 4.799 | 1.467 |
| | 48 | **1.553** | **0.815** | 1.673 | 0.854 | 2.130 | 1.024 | 2.505 | 1.041 | 3.024 | 1.145 | 5.004 | 1.542 | 7.551 | 2.057 | 4.800 | 1.468 |
| | 60 | **1.470** | **0.788** | 1.529 | 0.862 | 2.368 | 1.096 | 2.742 | 1.122 | 2.761 | 1.114 | 5.071 | 1.543 | 7.662 | 2.100 | 5.278 | 1.560 |
| ETTh1 | 96 | **0.370** | **0.400** | 0.375 | 0.399 | 0.375 | 0.399 | 0.376 | 0.415 | 0.435 | 0.446 | 0.941 | 0.769 | 0.664 | 0.612 | 0.878 | 0.740 |
| | 192 | 0.413 | 0.429 | 0.414 | 0.421 | 0.405 | 0.416 | 0.423 | 0.446 | 0.456 | 0.457 | 1.007 | 0.786 | 0.790 | 0.681 | 1.037 | 0.824 |
| | 336 | **0.422** | 0.440 | 0.431 | 0.436 | 0.439 | 0.443 | 0.444 | 0.462 | 0.486 | 0.487 | 1.038 | 0.784 | 0.891 | 0.738 | 1.238 | 0.932 |
| | 720 | **0.447** | 0.468 | 0.449 | 0.466 | 0.472 | 0.490 | 0.469 | 0.492 | 0.515 | 0.517 | 1.144 | 0.857 | 0.963 | 0.782 | 1.135 | 0.852 |
| ETTh2 | 96 | **0.274** | 0.337 | **0.274** | 0.336 | 0.289 | 0.353 | 0.332 | 0.374 | 0.332 | 0.368 | 1.549 | 0.952 | 0.645 | 0.597 | 2.116 | 1.197 |
| | 192 | 0.341 | 0.382 | 0.339 | 0.379 | 0.383 | 0.418 | 0.407 | 0.446 | 0.426 | 0.434 | 3.792 | 1.542 | 0.788 | 0.683 | 4.315 | 1.635 |
| | 336 | **0.329** | 0.384 | 0.331 | 0.380 | 0.448 | 0.465 | 0.400 | 0.447 | 0.477 | 0.479 | 4.215 | 1.642 | 0.907 | 0.747 | 1.124 | 1.604 |
| | 720 | **0.379** | **0.422** | 0.379 | 0.422 | 0.605 | 0.551 | 0.412 | 0.469 | 0.453 | 0.490 | 3.656 | 1.619 | 0.963 | 0.783 | 3.188 | 1.540 |
| ETTm1 | 96 | 0.293 | 0.346 | **0.290** | 0.342 | 0.299 | 0.343 | 0.326 | 0.390 | 0.510 | 0.492 | 0.626 | 0.560 | 0.543 | 0.510 | 0.600 | 0.546 |
| | 192 | 0.333 | 0.370 | **0.332** | 0.369 | 0.335 | 0.365 | 0.365 | 0.415 | 0.514 | 0.495 | 0.725 | 0.619 | 0.557 | 0.537 | 0.837 | 0.700 |
| | 336 | 0.369 | 0.392 | **0.366** | 0.392 | 0.369 | 0.386 | 0.392 | 0.425 | 0.510 | 0.492 | 1.005 | 0.741 | 0.754 | 0.655 | 1.124 | 0.832 |
| | 720 | **0.416** | **0.420** | 0.420 | 0.424 | 0.425 | 0.421 | 0.446 | 0.458 | 0.527 | 0.493 | 1.133 | 0.845 | 0.908 | 0.724 | 1.153 | 0.820 |
| ETTm2 | 96 | 0.166 | 0.256 | **0.165** | **0.255** | 0.167 | 0.260 | 0.180 | 0.271 | 0.205 | 0.293 | 0.355 | 0.462 | 0.435 | 0.507 | 0.768 | 0.642 |
| | 192 | 0.223 | 0.296 | **0.220** | **0.292** | 0.224 | 0.303 | 0.252 | 0.318 | 0.278 | 0.336 | 0.595 | 0.586 | 0.730 | 0.673 | 0.989 | 0.757 |
| | 336 | **0.274** | **0.329** | 0.278 | 0.329 | 0.281 | 0.342 | 0.324 | 0.364 | 0.343 | 0.379 | 1.270 | 0.871 | 1.201 | 0.845 | 1.334 | 0.872 |
| | 720 | **0.362** | **0.385** | 0.367 | 0.385 | 0.397 | 0.421 | 0.410 | 0.420 | 0.414 | 0.419 | 3.001 | 1.267 | 3.625 | 1.451 | 3.048 | 1.328 |

Table 3: Multivariate long-term forecasting results with supervised PatchTST. We use prediction lengths $T \in \{24, 36, 48, 60\}$ for ILI dataset and $T \in \{96, 192, 336, 720\}$ for the others. The best results are in **bold** and the second best are underlined.

| Models | | PatchTST | | | | | | DLinear | | FEDformer | | Autoformer | | Informer | |
|---|---|---|---|---|---|---|---|---|---|---|---|---|---|---|---|
| | | Fine-tuning | | Lin. Prob. | | Sup. | | | | | | | | | |
| Metric | | MSE | MAE | MSE | MAE | MSE | MAE | MSE | MAE | MSE | MAE | MSE | MAE | MSE | MAE |
| Weather | 96 | **0.144** | **0.193** | 0.158 | 0.209 | 0.152 | 0.199 | 0.176 | 0.237 | 0.238 | 0.314 | 0.249 | 0.329 | 0.354 | 0.405 |
| | 192 | **0.190** | **0.236** | 0.203 | 0.249 | 0.197 | 0.243 | 0.220 | 0.282 | 0.275 | 0.329 | 0.325 | 0.370 | 0.419 | 0.434 |
| | 336 | **0.244** | **0.280** | 0.251 | 0.285 | 0.249 | 0.283 | 0.265 | 0.319 | 0.339 | 0.377 | 0.351 | 0.391 | 0.583 | 0.543 |
| | 720 | **0.320** | **0.335** | 0.321 | 0.336 | 0.320 | 0.335 | 0.323 | 0.362 | 0.389 | 0.409 | 0.415 | 0.426 | 0.916 | 0.705 |
| Traffic | 96 | **0.352** | **0.244** | 0.399 | 0.294 | 0.367 | 0.251 | 0.410 | 0.282 | 0.576 | 0.359 | 0.597 | 0.371 | 0.733 | 0.410 |
| | 192 | **0.371** | **0.253** | 0.412 | 0.298 | 0.385 | 0.259 | 0.423 | 0.287 | 0.610 | 0.380 | 0.607 | 0.382 | 0.777 | 0.435 |
| | 336 | **0.381** | **0.257** | 0.425 | 0.306 | 0.398 | 0.265 | 0.436 | 0.296 | 0.608 | 0.375 | 0.623 | 0.387 | 0.776 | 0.434 |
| | 720 | **0.425** | **0.282** | 0.460 | 0.323 | 0.434 | 0.287 | 0.466 | 0.315 | 0.621 | 0.375 | 0.639 | 0.395 | 0.827 | 0.466 |
| Electricity | 96 | **0.126** | **0.221** | 0.138 | 0.237 | 0.130 | 0.222 | 0.140 | 0.237 | 0.186 | 0.302 | 0.196 | 0.313 | 0.304 | 0.393 |
| | 192 | **0.145** | **0.238** | 0.156 | 0.252 | 0.148 | 0.240 | 0.153 | 0.249 | 0.197 | 0.311 | 0.211 | 0.324 | 0.327 | 0.417 |
| | 336 | **0.164** | **0.256** | 0.170 | 0.265 | 0.167 | 0.261 | 0.169 | 0.267 | 0.213 | 0.328 | 0.214 | 0.327 | 0.333 | 0.422 |
| | 720 | **0.193** | **0.291** | 0.208 | 0.297 | 0.202 | 0.291 | 0.203 | 0.301 | 0.233 | 0.344 | 0.236 | 0.342 | 0.351 | 0.427 |

Table 4: Multivariate long-term forecasting results with self-supervised PatchTST. We use prediction lengths $T \in \{96, 192, 336, 720\}$. The best results are in **bold** and the second best are underlined.

**Comparison with Supervised Methods.** Table 4 compares the performance of PatchTST (with fine-tuning, linear probing, and supervising from scratch) with other supervised method. As shown in the table, on large datasets our pre-training procedure contributes a clear improvement compared to supervised training from scratch. By just fine-tuning the model head (linear probing), the forecasting performance is already comparable with training the entire network from scratch and better than DLinear. The best results are observed with end-to-end fine-tuning. Self-supervised PatchTST significantly outperforms other Transformer-based models on all the datasets.

**Transfer Learning.** We test the capability of transfering the pre-trained model to downstream tasks. In particular, we pre-train the model on Electricity dataset and fine-tune on other datasets. We observe from Table 5 that overall the fine-tuning MSE is lightly worse than pre-training and fine-tuning on the same dataset, which is reasonable. The fine-tuning performance is also worse than supervised training in some cases. However, the forecasting performance is still better than other

| Models | | PatchTST | | | | | DLinear | | FEDformer | | Autoformer | | Informer | |
|---|---|---|---|---|---|---|---|---|---|---|---|---|---|---|
| | | Fine-tuning | | Lin. Prob. | | Sup. | | | | | | | | | |
| Metric | | MSE | MAE | MSE | MAE | MSE | MAE | MSE | MAE | MSE | MAE | MSE | MAE | MSE | MAE |
| Weather | 96 | **0.145** | **0.195** | 0.163 | 0.216 | 0.152 | 0.199 | 0.176 | 0.237 | 0.238 | 0.314 | 0.249 | 0.329 | 0.354 | 0.405 |
| | 192 | **0.193** | **0.243** | 0.205 | 0.252 | 0.197 | 0.243 | 0.220 | 0.282 | 0.275 | 0.329 | 0.325 | 0.370 | 0.419 | 0.434 |
| | 336 | **0.244** | **0.280** | 0.253 | 0.289 | 0.249 | 0.283 | 0.265 | 0.319 | 0.339 | 0.377 | 0.351 | 0.391 | 0.583 | 0.543 |
| | 720 | 0.321 | 0.337 | **0.320** | 0.336 | 0.320 | **0.335** | 0.323 | 0.362 | 0.389 | 0.409 | 0.415 | 0.426 | 0.916 | 0.705 |
| Traffic | 96 | 0.388 | 0.273 | 0.400 | 0.288 | **0.367** | **0.251** | 0.410 | 0.282 | 0.576 | 0.359 | 0.597 | 0.371 | 0.733 | 0.410 |
| | 192 | 0.400 | 0.277 | 0.412 | 0.293 | **0.385** | **0.259** | 0.423 | 0.287 | 0.610 | 0.380 | 0.607 | 0.382 | 0.777 | 0.435 |
| | 336 | 0.408 | 0.280 | 0.425 | 0.307 | **0.398** | **0.265** | 0.436 | 0.296 | 0.608 | 0.375 | 0.623 | 0.387 | 0.776 | 0.434 |
| | 720 | 0.447 | 0.310 | 0.457 | 0.317 | **0.434** | **0.287** | 0.466 | 0.315 | 0.621 | 0.375 | 0.639 | 0.395 | 0.827 | 0.466 |

Table 5: Transfer learning task: PatchTST is pre-trained on Electricity dataset and the model is transferred to other datasets. The best results are in **bold** and the second best are underlined.

| Models | | IMP. | PatchTST | | | | BTSF | | TS2Vec | | TNC | | TS-TCC | |
|---|---|---|---|---|---|---|---|---|---|---|---|---|---|---|
| | | | Transferred | | Self-supervised | | | | | | | | | |
| Metrics | | MSE | MSE | MAE | MSE | MAE | MSE | MAE | MSE | MAE | MSE | MAE | MSE | MAE |
| ETTh1 | 24 | 42.3% | **0.312** | **0.362** | 0.322 | 0.369 | 0.541 | 0.519 | 0.599 | 0.534 | 0.632 | 0.596 | 0.653 | 0.610 |
| | 48 | 44.7% | **0.339** | **0.378** | 0.354 | 0.385 | 0.613 | 0.524 | 0.629 | 0.555 | 0.705 | 0.688 | 0.720 | 0.693 |
| | 168 | 34.5% | 0.424 | 0.437 | **0.419** | **0.424** | 0.640 | 0.532 | 0.755 | 0.636 | 1.097 | 0.993 | 1.129 | 1.044 |
| | 336 | 48.5% | 0.472 | 0.472 | **0.445** | **0.446** | 0.864 | 0.689 | 0.907 | 0.717 | 1.454 | 0.919 | 1.492 | 1.076 |
| | 720 | 48.8% | 0.508 | 0.507 | **0.487** | **0.478** | 0.993 | 0.712 | 1.048 | 0.790 | 1.604 | 1.118 | 1.603 | 1.206 |

Table 6: Representation learning methods comparison. Column name *transferred* implies pre-training PatchTST on Traffic dataset and transferring the representation to ETTh1, while *self-supervised* implies both pre-training and linear probing on ETTh1. The best and second best results are in **bold** and underlined. IMP. denotes the improvement on best results of PatchTST compared to that of baselines, which is in the range of $34.5\%$ to $48.8\%$ on various prediction lengths.

models. Note that as opposed to supervised PatchTST where the entire model is trained for each prediction horizon, here we only retrain the linear head or the entire model for much fewer epochs, which results in significant computational time reduction.

**Comparison with Other Self-supervised Methods.** We compare our self-supervised model with BTSF (Yang & Hong, 2022), TS2Vec (Yue et al., 2022), TNC (Tonekaboni et al., 2021), and TS-TCC (Eldele et al., 2021) which are among the state-of-the-art contrastive learning representation methods for time series [2]. We test the forecasting performance on ETTh1 dataset, where we only apply linear probing after the learned representation is obtained (only fine-tune the last linear layer) to make the comparison fair. Results from Table 6 strongly indicates the superior performance of PatchTST, both from pre-training on its own ETTh1 data (self-supervised columns) or pre-training on Traffic (transferred columns).

## 4.3 ABLATION STUDY

**Patching and Channel-independence.** We study the effects of patching and channel-independence in Table 7. We include FEDformer as the SOTA benchmark for Transformer-based model. By comparing results with and without the design of patching / channel-independence accordingly, one can observe that both of them are important factors in improving the forecasting performance. The motivation of patching is natural; furthermore this technique improves the running time and memory consumption as shown in Table 1 due to shorter Transformer sequence input. Channel-independence, on the other hand, may not be as intuitive as patching is in terms of the technical advantages. Therefore, we provide an in-depth analysis on the key factors that make channel-independence more preferable in Appendix A.7. More ablation study results are available in Appendix A.4.

**Varying Look-back Window.** In principle, a longer look-back window increases the receptive field, which will potentially improves the forecasting performance. However, as argued in (Zeng et al., 2022), this phenomenon hasn't been observed in most of the Transformer-based models. We also demonstrate in Figure 2 that in most cases, these Transformer-based baselines have not benefited from longer look-back window $L$, which indicates their ineffectiveness in capturing temporal information. In contrast, our PatchTST consistently reduces the MSE scores as the receptive field increases, which confirms our model's capability to learn from longer look-back window.

---

[2]We cite results of TS2Vec from (Yue et al., 2022) and {BTSF,TNC,TS-TCC} from (Yang & Hong, 2022).

| Models | | PatchTST | | | | | | | | FEDformer | |
|---|---|---|---|---|---|---|---|---|---|---|---|
| | | P+CI | | CI | | P | | Original | | | |
| Metric | | MSE | MAE | MSE | MAE | MSE | MAE | MSE | MAE | MSE | MAE |
| Weather | 96 | **0.152** | **0.199** | 0.164 | 0.213 | 0.168 | 0.223 | 0.177 | 0.236 | 0.238 | 0.314 |
| | 192 | **0.197** | **0.243** | 0.205 | 0.250 | 0.213 | 0.262 | 0.221 | 0.270 | 0.275 | 0.329 |
| | 336 | **0.249** | **0.283** | 0.255 | 0.289 | 0.266 | 0.300 | 0.271 | 0.306 | 0.339 | 0.377 |
| | 720 | **0.320** | **0.335** | 0.327 | 0.343 | 0.351 | 0.359 | 0.340 | 0.353 | 0.389 | 0.409 |
| Traffic | 96 | **0.367** | **0.251** | 0.397 | 0.271 | 0.595 | 0.376 | - | - | 0.576 | 0.359 |
| | 192 | **0.385** | **0.259** | 0.411 | 0.276 | 0.612 | 0.387 | - | - | 0.610 | 0.380 |
| | 336 | **0.398** | **0.265** | 0.423 | 0.282 | 0.651 | 0.391 | - | - | 0.608 | 0.375 |
| | 720 | **0.434** | **0.287** | 0.457 | 0.309 | - | - | - | - | 0.621 | 0.375 |
| Electricity | 96 | **0.130** | **0.222** | 0.136 | 0.231 | 0.196 | 0.307 | 0.205 | 0.318 | 0.186 | 0.302 |
| | 192 | **0.148** | **0.240** | 0.164 | 0.263 | 0.215 | 0.323 | - | - | 0.197 | 0.311 |
| | 336 | **0.167** | **0.261** | 0.168 | 0.262 | 0.228 | 0.338 | - | - | 0.213 | 0.328 |
| | 720 | **0.202** | **0.291** | 0.219 | 0.312 | 0.244 | 0.345 | - | - | 0.233 | 0.344 |

Table 7: Ablation study of patching and channel-independence in PatchTST. 4 cases are included: (a) both patching and channel-independence are included in model (P+CI); (b) only channel-independence (CI); (c) only patching (P); (d) neither of them is included (Original TST model). PatchTST means supervised PatchTST/42. '-' in table means the model runs out of GPU memory (NVIDIA A40 48GB) even with batch size 1. The best results are in **bold**.

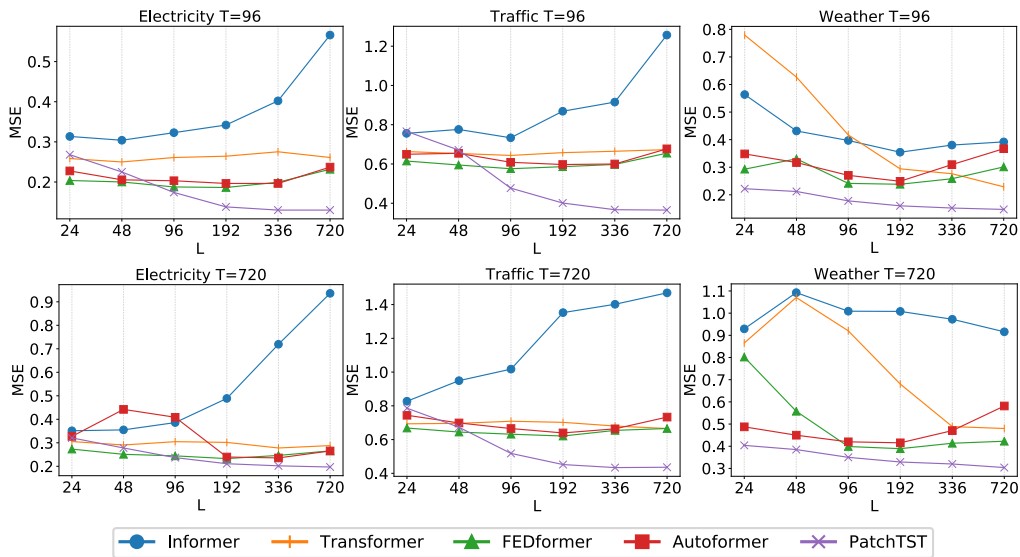

Figure 2: Forecasting performance (MSE) with varying look-back windows on 3 large datasets: Electricity, Traffic, and Weather. The look-back windows are selected to be $L = 24, 48, 96, 192, 336, 720$, and the prediction horizons are $T = 96, 720$. We use supervised PatchTST/42 and other open-source Transformer-based baselines for this experiment.

## 5  CONCLUSION AND FUTURE WORK

This paper proposes an effective design of Transformer-based models for time series forecasting tasks by introducing two key components: patching and channel-independent structure. Compared to the previous works, it could capture local semantic information and benefit from longer look-back windows. We not only show that our model outperforms other baselines in supervised learning, but also prove its promising capability in self-supervised representation learning and transfer learning.

Our model exhibits the potential to be the based model for future work of Transformer-based forecasting and be a building block for time series foundation models. Patching is simple but proven to be an effective operator that can be transferred easily to other models. Channel-independence, on the other hand, can be further exploited to incorporate the correlation between different channels. It would be an important future step to model the cross-channel dependencies properly.

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

## A  APPENDIX

### A.1  EXPERIMENTAL DETAILS

#### A.1.1  DATASETS

We use 8 popular multivariate datasets provided in (Wu et al., 2021) for forecasting and representation learning. *Weather*[3] dataset collects 21 meteorological indicators in Germany, such as humidity and air temperature. *Traffic*[4] dataset records the road occupancy rates from different sensors on San Francisco freeways. *Electricity*[5] is a dataset that describes 321 customers' hourly electricity consumption. *ILI*[6] dataset collects the number of patients and influenza-like illness ratio in a weekly frequency. *ETT*[7] (Electricity Transformer Temperature) datasets are collected from two different electric transformers labeled with 1 and 2, and each of them contains 2 different resolutions (15 minutes and 1 hour) denoted with m and h. Thus, in total we have 4 ETT datasets: *ETTm1*, *ETTm2*, *ETTh1*, and *ETTh2*.

There is an additional *Exchange-rate*[8] dataset mentioned in the original paper, which is the daily exchange-rate of eight different countries. However, financial datasets generally have different properties compared to time series datasets in other fields, for example the predictability. It is well known that if a market is efficient, the best prediction for $x_t$ will be just $x_{t-1}$ (Fama, 1970). Rossi (2013) argues that the toughest benchmark for exchange-rate forecasting is a random walk without drift. Also, Zeng et al. (2022) shows that by simply repeating the last value in the look-back window, the MSE loss on exchange-rate dataset can outperform or be comparable to the best results. Therefore, we are prudent in containing it into our benchmark.

#### A.1.2  DETAILS OF BASELINE SETTINGS

The default look-back windows for different baseline models could be different. For Transformer-based models, the default look-back window is $L = 96$; and for DLinear, the default look-back window is $L = 336$. The reason of this difference is that Transformer-based baselines are easy to overfit when look-back window is long while DLinear tend to underfit. However, this can possibly lead to an under-estimation of the baselines. To address this issue, we re-run FEDformer, Autoformer and Informer by ourselves for six different look-back window $L \in \{24, 48, 96, 192, 336, 720\}$. And for each forecasting task (aka each different prediction length on each dataset), we choose the best one from those six results. Thus it could be a strong baseline.

The ILI dataset is much smaller than the other datasets, so a different set of parameters is applied (the default look-back windows of Transformer-based models and DLinear are $L = 36$ and $L = 104$ respectively; we run FEDformer, Autoformer and Informer for six different look-back window $L \in \{24, 36, 48, 60, 104, 144\}$ and choose the best results).

#### A.1.3  BASELINES FROM TRADITIONAL MODELS

Time series has been an ancient field of study, with many traditional models developed, for example the famous ARIMA model (Box & Jenkins, 1970). With the bloom of deep learning community, many new models were proposed for sequence modeling and time series forecasting before Transformer appears, such as LSTM (Hochreiter & Schmidhuber, 1997), TCN (Bai et al., 2018) and DeepAR (Salinas et al., 2020). However, they are demonstrated to be not as effective as Transformer-based models in long-term forecasting tasks (Zhou et al., 2021; Wu et al., 2021), thus we don't include them in our baselines.

---

[3]https://www.bgc-jena.mpg.de/wetter/

[4]https://pems.dot.ca.gov/

[5]https://archive.ics.uci.edu/ml/datasets/ElectricityLoadDiagrams20112014

[6]https://gis.cdc.gov/grasp/fluview/fluportaldashboard.html

[7]https://github.com/zhouhaoyi/ETDataset

[8]https://github.com/laiguokun/multivariate-time-series-data

### A.1.4 MODEL PARAMETERS

By default, PatchTST contains 3 encoder layers with head number $H = 16$ and dimension of latent space $D = 128$. The feed forward network in Transformer encoder block consists of 2 linear layers with GELU (Hendrycks & Gimpel, 2016) activation function: one projecting the hidden representation $D = 128$ to a new dimension $F = 256$, and another layer that project it back to $D = 128$. For very small datasets (ILI, ETTh1, ETTh2), a reduced size of parameters is used ($H = 4, D = 16$ and $F = 128$) to mitigate the possible overfitting. Dropout with probability 0.2 is applied in the encoders for all experiments. The code will be publicly available.

### A.1.5 IMPLEMENTATION DETAILS

Although PatchTST processes channels in parallel which has to make multiple copies of the Transformer's weights, the computation can be implemented efficiently and does not require any special operator. The batch of samples of $x \in \mathbb{R}^{M \times L}$ with size $B \times M \times L$ is passed through the patching operator to generate a $4D$ tensor of size $B \times M \times P \times N$ which represents a batch of $x_p^{(i)} \in \mathbb{R}^{P \times N}$ in $M$ series. By reshaping the tensor to form a $3D$ one of size $(B \cdot M) \times P \times N$, this batch can be consumed by any standard Transformer implementation.

We further argue that our proposed PatchTST contains additional benefits: The components in Transformer backbone module shown in Figure 1 can differ across different input series, for instance the embedding layers and head layers. Note that if the embedding layers are designed differently for each group of time series, the reshaping step will be applied after embedding. Besides, the number of variables in the multivariate time series during the training may not need to match the number of series for testing. This is especially beneficial for self-supervised pre-training where the pre-training data can have different number of variables from the fine-tuning data.

## A.2 VISUALIZATION

We visualize the long-term forecasting results of supervised PatchTST/42 and other baselines in Figure 3. Here, we predict 192 steps ahead on Weather and Eletricity datasets and 60 steps ahead on ILI dataset. PatchTST provides the best forecasting both in terms of scale and bias.

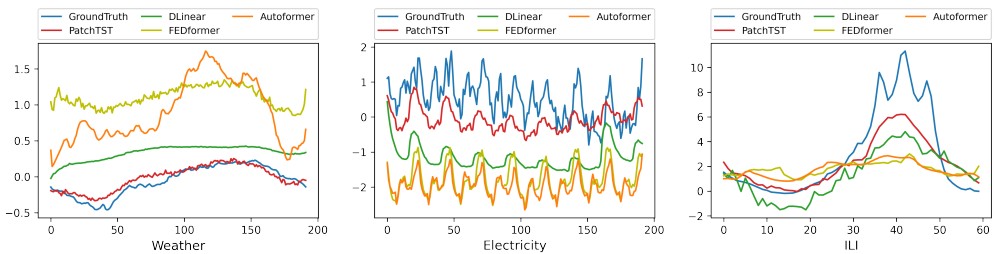

Figure 3: Visualization of 192-step forecasting on {Weather, Traffic} datasets with the look-back window $L = 336$ and 60-step forecasting on ILI dataset with $L = 104$. PatchTST (in red) can capture the trend and closest to the ground truth (in blue).

## A.3 UNIVARIATE FORECASTING

Table 8 summaries the results of univariate forecasting on ETT datasets. There is a target feature "oil temperature" within those datasets, which is the univariate series that we are trying to forecast. We cite the baseline results from (Zeng et al., 2022).

## A.4 MORE RESULTS ON ABLATION STUDY

### A.4.1 VARYING PATCH LENGTH

This experiment studies the effect of patch lengths to the forecasting performance. We fix the look-back window to be 336 and vary the patch lengths $P = [4, 8, 16, 24, 32, 40]$. The stride length is set

| Models | PatchTST/64 | | PatchTST/42 | | DLinear | | FEDformer | | Autoformer | | Informer | | LogTrans | |
|---|---|---|---|---|---|---|---|---|---|---|---|---|---|---|
| Metric | MSE | MAE | MSE | MAE | MSE | MAE | MSE | MAE | MSE | MAE | MSE | MAE | MSE | MAE |
| ETTh1 96 | 0.059 | 0.189 | **0.055** | **0.179** | 0.056 | 0.180 | 0.079 | 0.215 | 0.071 | 0.206 | 0.193 | 0.377 | 0.283 | 0.468 |
| ETTh1 192 | 0.074 | 0.215 | **0.071** | 0.205 | **0.071** | **0.204** | 0.104 | 0.245 | 0.114 | 0.262 | 0.217 | 0.395 | 0.234 | 0.409 |
| ETTh1 336 | **0.076** | **0.220** | 0.081 | 0.225 | 0.098 | 0.244 | 0.119 | 0.270 | 0.107 | 0.258 | 0.202 | 0.381 | 0.386 | 0.546 |
| ETTh1 720 | **0.087** | 0.236 | **0.087** | **0.232** | 0.189 | 0.359 | 0.142 | 0.299 | 0.126 | 0.283 | 0.183 | 0.355 | 0.475 | 0.629 |
| ETTh2 96 | 0.131 | 0.284 | **0.129** | 0.282 | 0.131 | **0.279** | 0.128 | 0.271 | 0.153 | 0.306 | 0.213 | 0.373 | 0.217 | 0.379 |
| ETTh2 192 | 0.171 | 0.329 | **0.168** | **0.328** | 0.176 | 0.329 | 0.185 | 0.330 | 0.204 | 0.351 | 0.227 | 0.387 | 0.281 | 0.429 |
| ETTh2 336 | **0.171** | **0.336** | 0.185 | 0.351 | 0.209 | 0.367 | 0.231 | 0.378 | 0.246 | 0.389 | 0.242 | 0.401 | 0.293 | 0.437 |
| ETTh2 720 | **0.223** | **0.380** | 0.224 | 0.383 | 0.276 | 0.426 | 0.278 | 0.420 | 0.268 | 0.409 | 0.291 | 0.439 | 0.218 | 0.387 |
| ETTm1 96 | **0.026** | 0.123 | **0.026** | **0.121** | 0.028 | 0.123 | 0.033 | 0.140 | 0.056 | 0.183 | 0.109 | 0.277 | 0.049 | 0.171 |
| ETTm1 192 | 0.040 | 0.151 | **0.039** | **0.150** | 0.045 | 0.156 | 0.058 | 0.186 | 0.081 | 0.216 | 0.151 | 0.310 | 0.157 | 0.317 |
| ETTm1 336 | **0.053** | 0.174 | **0.053** | **0.173** | 0.061 | 0.182 | 0.084 | 0.231 | 0.076 | 0.218 | 0.427 | 0.591 | 0.289 | 0.459 |
| ETTm1 720 | **0.073** | **0.206** | 0.074 | 0.207 | 0.080 | 0.210 | 0.102 | 0.250 | 0.110 | 0.267 | 0.438 | 0.586 | 0.430 | 0.579 |
| ETTm2 96 | 0.065 | 0.187 | 0.065 | 0.186 | **0.063** | **0.183** | 0.067 | 0.198 | 0.065 | 0.189 | 0.088 | 0.225 | 0.075 | 0.208 |
| ETTm2 192 | 0.093 | 0.231 | 0.094 | 0.231 | **0.092** | **0.227** | 0.102 | 0.245 | 0.118 | 0.256 | 0.132 | 0.283 | 0.129 | 0.275 |
| ETTm2 336 | 0.121 | 0.266 | 0.120 | 0.265 | **0.119** | **0.261** | 0.130 | 0.279 | 0.154 | 0.305 | 0.180 | 0.336 | 0.154 | 0.302 |
| ETTm2 720 | **0.172** | 0.322 | 0.171 | 0.322 | 0.175 | **0.320** | 0.178 | 0.325 | 0.182 | 0.335 | 0.300 | 0.435 | 0.160 | 0.321 |

Table 8: Univariate long-term forecasting results with supervised PatchTST. ETT datasets are used with prediction lengths $T \in \{96, 192, 336, 720\}$. The best results are in **bold**.

the same as patch length, meaning no overlapping between patches. The model is trained to predict 96 steps. One observation from Figure 4 is that MSE scores don't vary significantly with different choices of $P$, which indicate the robustness of our model against the patch length hyperparameter. Overall, PatchTST benefits from increased patch length, not only in forecasting performance but also in the computational reduction. The ideal patch length may depend on the dataset, but $P$ between $\{8, 16\}$ seems to be general good numbers.

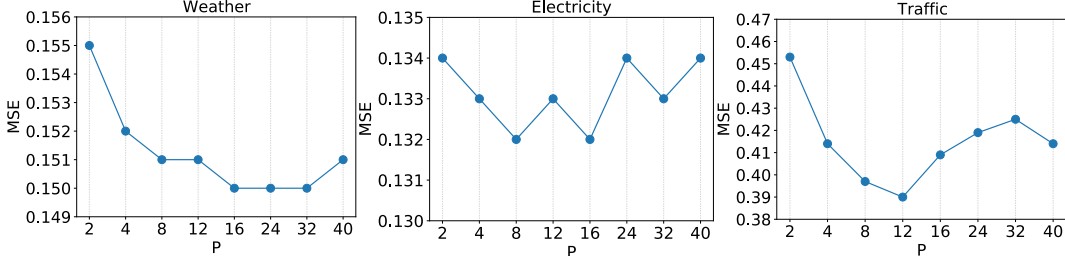

Figure 4: MSE scores with varying patch lengths $P = [2, 4, 8, 12, 16, 24, 32, 40]$ where the look-back window is 336 and the prediction length is 96.

### A.4.2 VARYING LOOK-BACK WINDOW

Here we provide a full benchmark of quantitative results in Table 9 for varying look-back window in supervised PatchTST/42 regarding Figure 2 in the main text. Generally speaking, our model gains performance improvement with increasing look-back window, which show the effectiveness of our model in learning information from longer receptive field.

### A.4.3 PATCHING AND CHANNEL-INDEPENDENCE

**Implementation Details.** For ablation study on patching and channel independence in section 4.3, we run different variants of PatchTST:

- Both patching and channel-independence are included in model (P+CI): this is the full PatchTST model that we have proposed in paper.

- Only channel-independence (CI): we simply set both patch length $P$ and stride $S$ to be 1 to avoid patching and only keep channel-independence.

- Only patching (P): referring to the implementation in A.1.5, instead of reshaping the $4D$ tensor from $B \times M \times P \times N$ to $(B \cdot M) \times P \times N$, we reshape it to $B \times (M \cdot P) \times N$ for channel-mixing with patching.

| $L$ | | 24(24) | | 48(36) | | 96(48) | | 192(60) | | 336(104) | | 720(144) | |
|---|---|---|---|---|---|---|---|---|---|---|---|---|---|
| Metric | | MSE | MAE | MSE | MAE | MSE | MAE | MSE | MAE | MSE | MAE | MSE | MAE |
| Weather | 96 | 0.222 | 0.246 | 0.212 | 0.243 | 0.178 | 0.219 | 0.160 | 0.204 | 0.152 | 0.199 | 0.147 | 0.198 |
| | 192 | 0.265 | 0.279 | 0.254 | 0.277 | 0.224 | 0.259 | 0.204 | 0.245 | 0.197 | 0.243 | 0.190 | 0.240 |
| | 336 | 0.325 | 0.322 | 0.310 | 0.316 | 0.278 | 0.298 | 0.257 | 0.285 | 0.249 | 0.283 | 0.242 | 0.282 |
| | 720 | 0.404 | 0.374 | 0.385 | 0.365 | 0.350 | 0.346 | 0.329 | 0.338 | 0.320 | 0.335 | 0.304 | 0.328 |
| Traffic | 96 | 0.766 | 0.419 | 0.671 | 0.381 | 0.477 | 0.305 | 0.401 | 0.267 | 0.367 | 0.251 | 0.365 | 0.251 |
| | 192 | 0.725 | 0.398 | 0.616 | 0.356 | 0.471 | 0.299 | 0.406 | 0.268 | 0.385 | 0.259 | 0.382 | 0.258 |
| | 336 | 0.752 | 0.410 | 0.635 | 0.364 | 0.485 | 0.305 | 0.421 | 0.277 | 0.398 | 0.265 | 0.398 | 0.267 |
| | 720 | 0.786 | 0.427 | 0.673 | 0.383 | 0.518 | 0.325 | 0.452 | 0.297 | 0.434 | 0.287 | 0.436 | 0.289 |
| Electricity | 96 | 0.268 | 0.316 | 0.225 | 0.293 | 0.174 | 0.259 | 0.138 | 0.230 | 0.130 | 0.222 | 0.130 | 0.224 |
| | 192 | 0.259 | 0.316 | 0.217 | 0.291 | 0.178 | 0.265 | 0.149 | 0.243 | 0.148 | 0.240 | 0.147 | 0.241 |
| | 336 | 0.283 | 0.335 | 0.238 | 0.309 | 0.196 | 0.282 | 0.169 | 0.262 | 0.167 | 0.261 | 0.163 | 0.259 |
| | 720 | 0.321 | 0.365 | 0.278 | 0.342 | 0.237 | 0.316 | 0.211 | 0.299 | 0.202 | 0.291 | 0.197 | 0.290 |
| ILI | 24 | 3.062 | 1.118 | 1.610 | 0.803 | 1.281 | 0.704 | 1.300 | 0.700 | 1.522 | 0.814 | 1.470 | 0.793 |
| | 36 | 2.732 | 1.071 | 1.262 | 0.731 | 1.251 | 0.752 | 1.367 | 0.776 | 1.430 | 0.834 | 1.518 | 0.856 |
| | 48 | 3.059 | 1.117 | 1.991 | 0.845 | 1.901 | 0.879 | 1.690 | 0.812 | 1.673 | 0.854 | 1.834 | 0.921 |
| | 60 | 2.610 | 1.057 | 1.702 | 0.829 | 1.611 | 0.844 | 1.526 | 0.795 | 1.529 | 0.862 | 1.656 | 0.885 |
| ETTh1 | 96 | 0.464 | 0.445 | 0.410 | 0.417 | 0.393 | 0.408 | 0.382 | 0.401 | 0.375 | 0.399 | 0.376 | 0.408 |
| | 192 | 0.521 | 0.474 | 0.469 | 0.448 | 0.445 | 0.434 | 0.428 | 0.425 | 0.414 | 0.421 | 0.413 | 0.431 |
| | 336 | 0.570 | 0.498 | 0.516 | 0.469 | 0.484 | 0.451 | 0.451 | 0.436 | 0.431 | 0.436 | 0.445 | 0.454 |
| | 720 | 0.575 | 0.522 | 0.509 | 0.487 | 0.480 | 0.471 | 0.452 | 0.459 | 0.449 | 0.466 | 0.458 | 0.477 |
| ETTh2 | 96 | 0.333 | 0.362 | 0.307 | 0.348 | 0.294 | 0.343 | 0.285 | 0.340 | 0.274 | 0.336 | 0.279 | 0.341 |
| | 192 | 0.422 | 0.409 | 0.397 | 0.399 | 0.377 | 0.393 | 0.356 | 0.386 | 0.339 | 0.379 | 0.349 | 0.386 |
| | 336 | 0.442 | 0.432 | 0.412 | 0.420 | 0.381 | 0.409 | 0.350 | 0.395 | 0.331 | 0.380 | 0.375 | 0.409 |
| | 720 | 0.462 | 0.453 | 0.434 | 0.441 | 0.412 | 0.433 | 0.395 | 0.427 | 0.379 | 0.422 | 0.394 | 0.434 |
| ETTm1 | 96 | 0.624 | 0.481 | 0.424 | 0.403 | 0.321 | 0.360 | 0.291 | 0.340 | 0.290 | 0.342 | 0.295 | 0.348 |
| | 192 | 0.671 | 0.507 | 0.468 | 0.429 | 0.362 | 0.384 | 0.328 | 0.365 | 0.332 | 0.369 | 0.334 | 0.373 |
| | 336 | 0.714 | 0.533 | 0.501 | 0.453 | 0.392 | 0.402 | 0.365 | 0.389 | 0.366 | 0.392 | 0.361 | 0.393 |
| | 720 | 0.744 | 0.554 | 0.553 | 0.484 | 0.450 | 0.435 | 0.422 | 0.423 | 0.420 | 0.424 | 0.416 | 0.419 |
| ETTm2 | 96 | 0.212 | 0.290 | 0.189 | 0.272 | 0.178 | 0.260 | 0.169 | 0.254 | 0.165 | 0.255 | 0.162 | 0.254 |
| | 192 | 0.282 | 0.334 | 0.260 | 0.317 | 0.249 | 0.307 | 0.230 | 0.294 | 0.220 | 0.292 | 0.216 | 0.293 |
| | 336 | 0.354 | 0.376 | 0.328 | 0.359 | 0.313 | 0.346 | 0.280 | 0.329 | 0.278 | 0.329 | 0.269 | 0.329 |
| | 720 | 0.458 | 0.433 | 0.429 | 0.415 | 0.400 | 0.398 | 0.378 | 0.386 | 0.367 | 0.385 | 0.350 | 0.380 |

Table 9: Multivariate long-term forecasting results with varying look-back window $L$ in supervised PatchTST/42.

- Neither patching nor channel-independence is included (Original), which is just the original TST model Zerveas et al. (2021).

Note that the default number of maximum epochs for Electricity and Traffic datasets are reduced from 100 to 20 for ablation experiments due to the huge space and time complexity of the original time series Transformer and channel independent model (no patching).

**Full Benchmark of Ablation Study.** The full benchmark is shown in Table 10, which is a completed version of Table 7 in the main text. We can observe that patching together with channel-independence achieves the best results from the table, especially on larger datasets (Weather, Traffic, and Electricity) where the models are less susceptible to overfitting and thus the results would be more convincing. As one can see the improvement is robust on both patching and channel-independence. It is interesting to see that the improvement on ILI dataset is significant as well.

### A.4.4 INSTANCE NORMALIZATION

Normalization is a technology used in many time series model to improve the forecasting performance (Kim et al., 2022; Chen et al., 2022; Zeng et al., 2022). In this experiment, we perform analysis on the effect of the instance normalization in our model. We train two models PatchTST/64 and PatchTST/42 with and without using instance normalization and observe the evaluated scores. As indicated in Table 11, instance normalization improves the forecasting performance slightly on two models. However, even without instance normalization operator, PatchTST still outperforms other Transformer methods on most of the datasets. This is to confirm that the improvement mainly comes from patching and channel independence designs.

### A.5 MORE RESULTS ON SELF-SUPERVISED REPRESENTATION LEARNING

### A.5.1 FULL BENCHMARK OF MULTIVARIATE FORECASTING

In this section we provide a full benchmark of multivariate forecasting results with self-supervised PatchTST in Table 12, which is an extended version of Table 4.

| Models | | PatchTST | | | | | | | | FEDformer | |
|---|---|---|---|---|---|---|---|---|---|---|---|
| | | P+CI | | CI | | P | | Original | | | |
| Metric | | MSE | MAE | MSE | MAE | MSE | MAE | MSE | MAE | MSE | MAE |
| Weather | 96 | **0.152** | **0.199** | 0.164 | 0.213 | 0.168 | 0.223 | 0.177 | 0.236 | 0.238 | 0.314 |
| | 192 | **0.197** | **0.243** | 0.205 | 0.250 | 0.213 | 0.262 | 0.221 | 0.270 | 0.275 | 0.329 |
| | 336 | **0.249** | **0.283** | 0.255 | 0.289 | 0.266 | 0.300 | 0.271 | 0.306 | 0.339 | 0.377 |
| | 720 | **0.320** | **0.335** | 0.327 | 0.343 | 0.351 | 0.359 | 0.340 | 0.353 | 0.389 | 0.409 |
| Traffic | 96 | **0.367** | **0.251** | 0.397 | 0.271 | 0.595 | 0.376 | - | - | 0.576 | 0.359 |
| | 192 | **0.385** | **0.259** | 0.411 | 0.276 | 0.612 | 0.387 | - | - | 0.610 | 0.380 |
| | 336 | **0.398** | **0.265** | 0.423 | 0.282 | 0.651 | 0.391 | - | - | 0.608 | 0.375 |
| | 720 | **0.434** | **0.287** | 0.457 | 0.309 | - | - | - | - | 0.621 | 0.375 |
| Electricity | 96 | **0.130** | **0.222** | 0.136 | 0.231 | 0.196 | 0.307 | 0.205 | 0.318 | 0.186 | 0.302 |
| | 192 | **0.148** | **0.240** | 0.164 | 0.263 | 0.215 | 0.323 | - | - | 0.197 | 0.311 |
| | 336 | **0.167** | **0.261** | 0.168 | 0.262 | 0.228 | 0.338 | - | - | 0.213 | 0.328 |
| | 720 | **0.202** | **0.291** | 0.219 | 0.312 | 0.244 | 0.345 | - | - | 0.233 | 0.344 |
| ILI | 24 | **1.522** | **0.814** | 2.111 | 1.048 | 2.157 | 0.964 | 2.737 | 1.081 | 2.624 | 1.095 |
| | 36 | **1.430** | **0.834** | 2.000 | 1.002 | 2.564 | 1.058 | 2.126 | 0.935 | 2.516 | 1.021 |
| | 48 | **1.673** | **0.854** | 2.167 | 1.029 | 2.348 | 1.022 | 2.178 | 0.971 | 2.505 | 1.041 |
| | 60 | **1.529** | **0.862** | 2.075 | 1.021 | 2.486 | 1.065 | 2.354 | 1.026 | 2.742 | 1.122 |
| ETTh1 | 96 | 0.375 | 0.399 | **0.365** | **0.395** | 0.416 | 0.438 | 0.455 | 0.459 | 0.376 | 0.415 |
| | 192 | 0.414 | 0.421 | **0.403** | **0.415** | 0.459 | 0.464 | 0.503 | 0.486 | 0.423 | 0.446 |
| | 336 | 0.431 | 0.436 | **0.430** | **0.433** | 0.484 | 0.480 | 0.514 | 0.503 | 0.444 | 0.462 |
| | 720 | **0.449** | 0.466 | **0.449** | **0.454** | 0.500 | 0.494 | 0.531 | 0.520 | 0.469 | 0.492 |
| ETTh2 | 96 | **0.274** | **0.336** | 0.277 | 0.337 | 0.334 | 0.388 | 0.348 | 0.394 | 0.332 | 0.374 |
| | 192 | **0.339** | **0.379** | 0.343 | 0.384 | 0.381 | 0.418 | 0.395 | 0.424 | 0.407 | 0.446 |
| | 336 | **0.331** | **0.380** | 0.333 | 0.383 | 0.361 | 0.414 | 0.369 | 0.419 | 0.400 | 0.447 |
| | 720 | **0.379** | 0.422 | **0.379** | **0.420** | 0.423 | 0.448 | 0.433 | 0.458 | 0.412 | 0.469 |
| ETTm1 | 96 | **0.290** | **0.342** | 0.300 | 0.354 | 0.326 | 0.368 | 0.324 | 0.370 | 0.326 | 0.390 |
| | 192 | **0.332** | **0.369** | 0.333 | 0.374 | 0.391 | 0.405 | 0.373 | 0.398 | 0.365 | 0.415 |
| | 336 | **0.366** | **0.392** | 0.369 | 0.397 | 0.427 | 0.425 | 0.415 | 0.421 | 0.392 | 0.425 |
| | 720 | 0.420 | 0.424 | **0.413** | **0.423** | 0.481 | 0.457 | 0.480 | 0.459 | 0.446 | 0.458 |
| ETTm2 | 96 | **0.165** | **0.255** | 0.166 | 0.259 | 0.195 | 0.274 | 0.208 | 0.289 | 0.180 | 0.271 |
| | 192 | **0.220** | **0.292** | 0.223 | 0.295 | 0.259 | 0.314 | 0.265 | 0.328 | 0.252 | 0.318 |
| | 336 | **0.278** | **0.329** | 0.279 | 0.330 | 0.297 | 0.345 | 0.323 | 0.365 | 0.324 | 0.364 |
| | 720 | **0.367** | **0.385** | 0.370 | 0.387 | 0.400 | 0.404 | 0.469 | 0.444 | 0.410 | 0.420 |

Table 10: Ablation study of patching (P) and channel-independence (CI) in PatchTST/42. A full benchmark regarding Table 7. The best results are in **bold**. '-' in table means the model runs out of GPU memory (NVIDIA A40 48GB) even with batch size 1.

| Models | PatchTST/64 (+in) | | PatchTST/64 (-in) | | PatchTST/42 (+in) | | PatchTST/42 (-in) | | FEDformer | | Autoformer | | Informer | |
|---|---|---|---|---|---|---|---|---|---|---|---|---|---|---|
| Metric | MSE | MAE | MSE | MAE | MSE | MAE | MSE | MAE | MSE | MAE | MSE | MAE | MSE | MAE |
| Weather 96 | **0.149** | **0.198** | 0.161 | 0.219 | _0.152_ | _0.199_ | 0.156 | 0.210 | 0.238 | 0.314 | 0.249 | 0.329 | 0.354 | 0.405 |
| 192 | **0.194** | **0.241** | 0.201 | 0.254 | _0.197_ | _0.243_ | 0.199 | 0.250 | 0.275 | 0.329 | 0.325 | 0.370 | 0.419 | 0.434 |
| 336 | **0.245** | **0.282** | 0.253 | 0.298 | _0.249_ | _0.283_ | 0.248 | 0.294 | 0.339 | 0.377 | 0.351 | 0.391 | 0.583 | 0.543 |
| 720 | **0.314** | **0.334** | 0.323 | 0.357 | _0.320_ | _0.335_ | 0.313 | 0.342 | 0.389 | 0.409 | 0.415 | 0.426 | 0.916 | 0.705 |
| Traffic 96 | **0.360** | **0.249** | 0.413 | 0.295 | _0.367_ | _0.251_ | 0.425 | 0.299 | 0.576 | 0.359 | 0.597 | 0.371 | 0.733 | 0.410 |
| 192 | **0.379** | **0.256** | 0.425 | 0.302 | _0.385_ | _0.259_ | 0.439 | 0.302 | 0.610 | 0.380 | 0.607 | 0.382 | 0.777 | 0.435 |
| 336 | **0.392** | **0.264** | 0.435 | 0.307 | _0.398_ | _0.265_ | 0.456 | 0.316 | 0.608 | 0.375 | 0.623 | 0.387 | 0.776 | 0.434 |
| 720 | **0.432** | **0.286** | 0.473 | 0.321 | _0.434_ | _0.287_ | 0.488 | 0.333 | 0.621 | 0.375 | 0.639 | 0.395 | 0.827 | 0.466 |
| Electricity 96 | **0.129** | **0.222** | 0.133 | 0.230 | _0.130_ | **0.222** | 0.131 | 0.226 | 0.186 | 0.302 | 0.196 | 0.313 | 0.304 | 0.393 |
| 192 | **0.147** | **0.240** | 0.148 | 0.244 | _0.148_ | **0.240** | 0.150 | 0.244 | 0.197 | 0.311 | 0.211 | 0.324 | 0.327 | 0.417 |
| 336 | **0.163** | **0.259** | 0.164 | 0.262 | _0.167_ | _0.261_ | 0.168 | 0.267 | 0.213 | 0.328 | 0.214 | 0.327 | 0.333 | 0.422 |
| 720 | **0.197** | **0.290** | 0.196 | 0.291 | _0.202_ | 0.291 | 0.201 | 0.298 | 0.233 | 0.344 | 0.236 | 0.342 | 0.351 | 0.427 |
| ILI 24 | **1.319** | **0.754** | 3.563 | 1.317 | _1.522_ | _0.814_ | 3.489 | 1.345 | 2.624 | 1.095 | 2.906 | 1.182 | 4.657 | 1.449 |
| 36 | _1.579_ | _0.870_ | 3.426 | 1.205 | **1.430** | **0.834** | 4.629 | 1.550 | 2.516 | 1.021 | 2.585 | 1.038 | 4.650 | 1.463 |
| 48 | **1.553** | **0.815** | 4.309 | 1.449 | _1.673_ | _0.854_ | 3.746 | 1.383 | 2.505 | 1.041 | 3.024 | 1.145 | 5.004 | 1.542 |
| 60 | **1.470** | **0.788** | 4.065 | 1.402 | _1.529_ | _0.862_ | 5.174 | 1.622 | 2.742 | 1.122 | 2.761 | 1.114 | 5.071 | 1.543 |
| ETTh1 96 | **0.370** | _0.400_ | 0.385 | 0.410 | _0.375_ | **0.399** | 0.388 | 0.412 | 0.376 | 0.415 | 0.435 | 0.446 | 0.941 | 0.769 |
| 192 | _0.413_ | 0.429 | 0.417 | 0.432 | 0.414 | _0.421_ | 0.430 | 0.438 | 0.423 | 0.446 | 0.456 | 0.457 | 1.007 | 0.786 |
| 336 | **0.422** | _0.440_ | 0.439 | 0.449 | _0.431_ | **0.436** | 0.454 | 0.458 | 0.444 | 0.462 | 0.486 | 0.487 | 1.038 | 0.784 |
| 720 | **0.447** | _0.468_ | 0.478 | 0.494 | _0.449_ | **0.466** | 0.494 | 0.497 | 0.469 | 0.492 | 0.515 | 0.517 | 1.144 | 0.857 |
| ETTh2 96 | **0.274** | _0.337_ | 0.299 | 0.359 | **0.274** | **0.336** | 0.313 | 0.374 | 0.332 | 0.374 | 0.332 | 0.368 | 1.549 | 0.952 |
| 192 | _0.341_ | _0.382_ | 0.354 | 0.404 | **0.339** | **0.379** | 0.402 | 0.432 | 0.407 | 0.446 | 0.426 | 0.434 | 3.792 | 1.542 |
| 336 | **0.329** | _0.384_ | 0.374 | 0.420 | _0.331_ | **0.380** | 0.448 | 0.465 | 0.400 | 0.447 | 0.477 | 0.479 | 4.215 | 1.642 |
| 720 | **0.379** | **0.422** | 0.479 | 0.492 | **0.379** | **0.422** | 0.688 | 0.588 | 0.412 | 0.469 | 0.453 | 0.490 | 3.656 | 1.619 |
| ETTm1 96 | _0.293_ | 0.346 | 0.308 | 0.358 | **0.290** | **0.342** | 0.308 | 0.358 | 0.326 | 0.390 | 0.510 | 0.492 | 0.626 | 0.560 |
| 192 | _0.333_ | 0.370 | 0.335 | 0.375 | **0.332** | _0.369_ | 0.356 | 0.390 | 0.365 | 0.415 | 0.514 | 0.495 | 0.725 | 0.619 |
| 336 | _0.369_ | _0.392_ | 0.362 | 0.392 | 0.366 | _0.392_ | 0.389 | 0.411 | 0.392 | 0.425 | 0.510 | 0.492 | 1.005 | 0.741 |
| 720 | **0.416** | **0.420** | 0.432 | 0.429 | _0.420_ | 0.424 | 0.430 | 0.439 | 0.446 | 0.458 | 0.527 | 0.493 | 1.133 | 0.845 |
| ETTm2 96 | _0.166_ | _0.256_ | 0.172 | 0.258 | **0.165** | **0.255** | 0.167 | 0.257 | 0.180 | 0.271 | 0.205 | 0.293 | 0.355 | 0.462 |
| 192 | _0.223_ | _0.296_ | 0.245 | 0.306 | **0.220** | **0.292** | 0.226 | 0.303 | 0.252 | 0.318 | 0.278 | 0.336 | 0.595 | 0.586 |
| 336 | **0.274** | **0.329** | 0.306 | 0.346 | _0.278_ | **0.329** | 0.301 | 0.348 | 0.324 | 0.364 | 0.343 | 0.379 | 1.270 | 0.871 |
| 720 | **0.362** | **0.385** | 0.391 | 0.404 | _0.367_ | **0.385** | 0.392 | 0.407 | 0.410 | 0.420 | 0.414 | 0.419 | 3.001 | 1.267 |

Table 11: Multivariate long-term forecasting results of supervised PatchTST with instance normalization (+in) or without instance normalization (-in). The best results are in **bold** and the second best are underlined. Although the models perform slightly better with instance normalization, compared to other Transformer models, the proposed approach achieve significantly better forecasting on most of the datasets even without instance normalization.

### A.5.2 FULL BENCHMARK OF TRANSFER LEARNING

In this section we provide Table 13 which contains the results of pre-training on Electricity dataset then transferred to other 6 datasets. Except Traffic data, the number of time series employed in the pre-training is much larger than the number of series during fine-tuning. It is a full version with respect to Table 5 in the main text and more cogently proves the capability to do transfer learning using our PatchTST model.

## A.6 ROBUSTNESS ANALYSIS

### A.6.1 RESULTS WITH DIFFERENT RANDOM SEEDS

The results reported in the main text and appendix above are run with the fixed random seed 2021. To examine the robustness of our results, we train the supervised PatchTST model with 5 different random seeds: 2019, 2020, 2021, 2022, 2023 and calculate the MSE and MAE scores with each selected seed. The mean and standard derivation of the results are reported in Table 14. It is clear that the variances are considerably small which indicates the robustness against choice of random seeds of our model.

We also validate the self-supervised PatchTST model on different runs. We pre-train the model once and fine-tune the model 5 times with different random batch selections. The mean and standard derivation across different runs are also provided in Table 14. We also observe that the variance is insignificant especially on large datasets while higher variance can be seen on smaller datasets.

| Models | | PatchTST | | | | | | DLinear | | FEDformer | | Autoformer | | Informer | |
|---|---|---|---|---|---|---|---|---|---|---|---|---|---|---|---|
| | | Fine-tuning | | Lin. Prob. | | Sup. | | | | | | | | | |
| Metric | | MSE | MAE | MSE | MAE | MSE | MAE | MSE | MAE | MSE | MAE | MSE | MAE | MSE | MAE |
| Weather | 96 | **0.144** | **0.193** | 0.158 | 0.209 | 0.152 | 0.199 | 0.176 | 0.237 | 0.238 | 0.314 | 0.249 | 0.329 | 0.354 | 0.405 |
| | 192 | **0.190** | **0.236** | 0.203 | 0.249 | 0.197 | 0.243 | 0.220 | 0.282 | 0.275 | 0.329 | 0.325 | 0.370 | 0.419 | 0.434 |
| | 336 | **0.244** | **0.280** | 0.251 | 0.285 | 0.249 | 0.283 | 0.265 | 0.319 | 0.339 | 0.377 | 0.351 | 0.391 | 0.583 | 0.543 |
| | 720 | **0.320** | **0.335** | 0.321 | 0.336 | **0.320** | **0.335** | 0.323 | 0.362 | 0.389 | 0.409 | 0.415 | 0.426 | 0.916 | 0.705 |
| Traffic | 96 | **0.352** | **0.244** | 0.399 | 0.294 | 0.367 | 0.251 | 0.410 | 0.282 | 0.576 | 0.359 | 0.597 | 0.371 | 0.733 | 0.410 |
| | 192 | **0.371** | **0.253** | 0.412 | 0.298 | 0.385 | 0.259 | 0.423 | 0.287 | 0.610 | 0.380 | 0.607 | 0.382 | 0.777 | 0.435 |
| | 336 | **0.381** | **0.257** | 0.425 | 0.306 | 0.398 | 0.265 | 0.436 | 0.296 | 0.608 | 0.375 | 0.623 | 0.387 | 0.776 | 0.434 |
| | 720 | **0.425** | **0.282** | 0.460 | 0.323 | 0.434 | 0.287 | 0.466 | 0.315 | 0.621 | 0.375 | 0.639 | 0.395 | 0.827 | 0.466 |
| Electricity | 96 | **0.126** | **0.221** | 0.138 | 0.237 | 0.130 | 0.222 | 0.140 | 0.237 | 0.186 | 0.302 | 0.196 | 0.313 | 0.304 | 0.393 |
| | 192 | **0.145** | **0.238** | 0.156 | 0.252 | 0.148 | 0.240 | 0.153 | 0.249 | 0.197 | 0.311 | 0.211 | 0.324 | 0.327 | 0.417 |
| | 336 | **0.164** | **0.256** | 0.170 | 0.265 | 0.167 | 0.261 | 0.169 | 0.267 | 0.213 | 0.328 | 0.214 | 0.327 | 0.333 | 0.422 |
| | 720 | **0.193** | **0.291** | 0.208 | 0.297 | 0.202 | **0.291** | 0.203 | 0.301 | 0.233 | 0.344 | 0.236 | 0.342 | 0.351 | 0.427 |
| ETTh1 | 96 | **0.366** | **0.397** | 0.371 | 0.400 | 0.375 | 0.399 | 0.375 | 0.399 | 0.376 | 0.415 | 0.435 | 0.446 | 0.941 | 0.769 |
| | 192 | 0.431 | 0.443 | 0.411 | 0.428 | 0.414 | 0.421 | **0.405** | **0.416** | 0.423 | 0.446 | 0.456 | 0.457 | 1.007 | 0.786 |
| | 336 | 0.450 | 0.456 | 0.445 | 0.446 | **0.431** | **0.436** | 0.439 | 0.443 | 0.444 | 0.462 | 0.486 | 0.487 | 1.038 | 0.784 |
| | 720 | 0.472 | 0.484 | 0.487 | 0.478 | **0.449** | **0.466** | 0.472 | 0.490 | 0.469 | 0.492 | 0.515 | 0.517 | 1.144 | 0.857 |
| ETTh2 | 96 | 0.284 | 0.343 | 0.285 | 0.344 | **0.274** | **0.336** | 0.289 | 0.353 | 0.332 | 0.374 | 0.332 | 0.368 | 1.549 | 0.952 |
| | 192 | 0.355 | 0.387 | 0.356 | 0.387 | **0.339** | **0.379** | 0.383 | 0.418 | 0.407 | 0.446 | 0.426 | 0.434 | 3.792 | 1.542 |
| | 336 | 0.379 | 0.411 | 0.377 | 0.410 | **0.331** | **0.380** | 0.448 | 0.465 | 0.400 | 0.447 | 0.477 | 0.479 | 4.215 | 1.642 |
| | 720 | 0.400 | 0.435 | 0.395 | 0.434 | **0.379** | **0.422** | 0.605 | 0.551 | 0.412 | 0.469 | 0.453 | 0.490 | 3.656 | 1.619 |
| ETTm1 | 96 | **0.289** | 0.344 | 0.292 | 0.348 | 0.290 | **0.342** | 0.299 | 0.343 | 0.326 | 0.390 | 0.510 | 0.492 | 0.626 | 0.560 |
| | 192 | **0.323** | 0.368 | 0.329 | 0.369 | 0.332 | 0.369 | 0.335 | **0.365** | 0.365 | 0.415 | 0.514 | 0.495 | 0.725 | 0.619 |
| | 336 | **0.353** | 0.387 | 0.364 | 0.391 | 0.366 | 0.392 | 0.369 | **0.386** | 0.392 | 0.425 | 0.510 | 0.492 | 1.005 | 0.741 |
| | 720 | **0.398** | **0.416** | 0.415 | 0.419 | 0.420 | 0.424 | 0.425 | 0.421 | 0.446 | 0.458 | 0.527 | 0.493 | 1.133 | 0.845 |
| ETTm2 | 96 | 0.166 | 0.256 | 0.167 | 0.257 | **0.165** | **0.255** | 0.167 | 0.260 | 0.180 | 0.271 | 0.205 | 0.293 | 0.355 | 0.462 |
| | 192 | 0.221 | 0.295 | 0.229 | 0.300 | **0.220** | **0.292** | 0.224 | 0.303 | 0.252 | 0.318 | 0.278 | 0.336 | 0.595 | 0.586 |
| | 336 | **0.278** | 0.333 | 0.289 | 0.343 | **0.278** | **0.329** | 0.281 | 0.342 | 0.324 | 0.364 | 0.343 | 0.379 | 1.270 | 0.871 |
| | 720 | **0.365** | 0.388 | 0.363 | 0.386 | 0.367 | **0.385** | 0.397 | 0.421 | 0.410 | 0.420 | 0.414 | 0.419 | 3.001 | 1.267 |

Table 12: Multivariate long-term forecasting results with self-supervised PatchTST. An full benchmark regarding Table 4. The best results are in **bold**.

| Models | | PatchTST | | | | | | DLinear | | FEDformer | | Autoformer | | Informer | |
|---|---|---|---|---|---|---|---|---|---|---|---|---|---|---|---|
| | | Fine-tuning | | Lin. Prob. | | Sup. | | | | | | | | | |
| Metric | | MSE | MAE | MSE | MAE | MSE | MAE | MSE | MAE | MSE | MAE | MSE | MAE | MSE | MAE |
| Weather | 96 | **0.145** | **0.195** | 0.163 | 0.216 | 0.152 | 0.199 | 0.176 | 0.237 | 0.238 | 0.314 | 0.249 | 0.329 | 0.354 | 0.405 |
| | 192 | **0.193** | **0.243** | 0.205 | 0.252 | 0.197 | **0.243** | 0.220 | 0.282 | 0.275 | 0.329 | 0.325 | 0.370 | 0.419 | 0.434 |
| | 336 | **0.244** | **0.280** | 0.253 | 0.289 | 0.249 | 0.283 | 0.265 | 0.319 | 0.339 | 0.377 | 0.351 | 0.391 | 0.583 | 0.543 |
| | 720 | 0.321 | 0.337 | **0.320** | 0.336 | **0.320** | **0.335** | 0.323 | 0.362 | 0.389 | 0.409 | 0.415 | 0.426 | 0.916 | 0.705 |
| Traffic | 96 | 0.388 | 0.273 | 0.400 | 0.288 | **0.367** | **0.251** | 0.410 | 0.282 | 0.576 | 0.359 | 0.597 | 0.371 | 0.733 | 0.410 |
| | 192 | 0.400 | 0.277 | 0.412 | 0.293 | **0.385** | **0.259** | 0.423 | 0.287 | 0.610 | 0.380 | 0.607 | 0.382 | 0.777 | 0.435 |
| | 336 | 0.408 | 0.280 | 0.425 | 0.307 | **0.398** | **0.265** | 0.436 | 0.296 | 0.608 | 0.375 | 0.623 | 0.387 | 0.776 | 0.434 |
| | 720 | 0.447 | 0.310 | 0.457 | 0.317 | **0.434** | **0.287** | 0.466 | 0.315 | 0.621 | 0.375 | 0.639 | 0.395 | 0.827 | 0.466 |
| ETTh1 | 96 | **0.368** | **0.398** | 0.372 | 0.402 | 0.375 | 0.399 | 0.375 | 0.399 | 0.376 | 0.415 | 0.435 | 0.446 | 0.941 | 0.769 |
| | 192 | 0.425 | 0.439 | 0.411 | 0.428 | 0.414 | 0.421 | **0.405** | **0.416** | 0.423 | 0.446 | 0.456 | 0.457 | 1.007 | 0.786 |
| | 336 | 0.470 | 0.471 | 0.442 | 0.454 | **0.431** | **0.436** | 0.439 | 0.443 | 0.444 | 0.462 | 0.486 | 0.487 | 1.038 | 0.784 |
| | 720 | 0.472 | 0.484 | 0.497 | 0.501 | **0.449** | **0.466** | 0.472 | 0.490 | 0.469 | 0.492 | 0.515 | 0.517 | 1.144 | 0.857 |
| ETTh2 | 96 | 0.285 | 0.345 | 0.280 | 0.341 | **0.274** | **0.334** | 0.289 | 0.353 | 0.332 | 0.374 | 0.332 | 0.368 | 1.549 | 0.952 |
| | 192 | 0.350 | 0.388 | 0.350 | 0.387 | **0.339** | **0.379** | 0.383 | 0.418 | 0.407 | 0.446 | 0.426 | 0.434 | 3.792 | 1.542 |
| | 336 | 0.378 | 0.410 | 0.373 | 0.410 | **0.331** | **0.380** | 0.448 | 0.465 | 0.400 | 0.447 | 0.477 | 0.479 | 4.215 | 1.642 |
| | 720 | 0.401 | 0.438 | 0.398 | 0.436 | **0.379** | **0.422** | 0.605 | 0.551 | 0.412 | 0.469 | 0.453 | 0.490 | 3.656 | 1.619 |
| ETTm1 | 96 | **0.288** | **0.345** | 0.291 | 0.346 | 0.290 | 0.342 | 0.299 | 0.343 | 0.326 | 0.390 | 0.510 | 0.492 | 0.626 | 0.560 |
| | 192 | **0.330** | 0.372 | 0.335 | 0.373 | 0.332 | 0.369 | 0.335 | **0.365** | 0.365 | 0.415 | 0.514 | 0.495 | 0.725 | 0.619 |
| | 336 | **0.359** | 0.392 | 0.365 | 0.391 | 0.366 | 0.392 | 0.369 | **0.386** | 0.392 | 0.425 | 0.510 | 0.492 | 1.005 | 0.741 |
| | 720 | **0.406** | **0.421** | 0.423 | 0.424 | 0.420 | 0.424 | 0.425 | 0.421 | 0.446 | 0.458 | 0.527 | 0.493 | 1.133 | 0.845 |
| ETTm2 | 96 | **0.164** | 0.256 | 0.166 | 0.257 | 0.165 | **0.255** | 0.167 | 0.260 | 0.180 | 0.271 | 0.205 | 0.293 | 0.355 | 0.462 |
| | 192 | 0.223 | 0.296 | 0.221 | 0.295 | **0.220** | **0.292** | 0.224 | 0.303 | 0.252 | 0.318 | 0.278 | 0.336 | 0.595 | 0.586 |
| | 336 | **0.277** | 0.332 | **0.277** | 0.332 | 0.278 | **0.329** | 0.281 | 0.342 | 0.324 | 0.364 | 0.343 | 0.379 | 1.270 | 0.871 |
| | 720 | **0.365** | 0.387 | 0.368 | 0.389 | 0.367 | **0.385** | 0.397 | 0.421 | 0.410 | 0.420 | 0.414 | 0.419 | 3.001 | 1.267 |

Table 13: Transfer learning task: PatchTST is pre-trained on Electricity dataset and the model is transferred to other datasets. A full benchmark regarding Table 13. The best results are in **bold**.

### A.6.2 RESULTS WITH DIFFERENT MODEL PARAMETERS

To see whether PatchTST is sensitive to the choice of Transformer settings, we perform another experiments with varying model parameters. We vary the number of Transformer layers $L = \{3, 4, 5\}$ and select the model dimension $D = \{128, 256\}$ while the inner-layer of the feed forward network is $F = 2D$. In total, there are 6 different sets of model hyper-parameters to examine. Figure 5 shows the MSE scores of these combinations on different datasets. Except ILI dataset reveals high variance with different hyper-parameter settings, other datasets are robust to the choice of model hyper-parameters.

| $L$ | | PatchTST/42 supervised | | PatchTST/42 self-supervised | |
|---|---|---|---|---|---|
| Metric | | MSE | MAE | MSE | MAE |
| Weather | 96 | 0.1525±0.0024 | 0.2002±0.0023 | **0.1450±0.0008** | **0.1937±0.0010** |
| | 192 | 0.1975±0.0015 | 0.2434±0.0010 | **0.1893±0.0003** | **0.2364±0.0006** |
| | 336 | 0.2494±0.0012 | 0.2841±0.0014 | **0.2413±0.0003** | **0.2774±0.0005** |
| | 720 | 0.3194±0.0002 | 0.3352±0.0003 | **0.3156±0.0020** | **0.3316±0.0016** |
| Traffic | 96 | 0.3669±0.0006 | 0.2504±0.0007 | **0.3528±0.0022** | **0.2443±0.0016** |
| | 192 | 0.3858±0.0004 | 0.2586±0.0004 | **0.3729±0.0013** | **0.2531±0.0009** |
| | 336 | 0.3994±0.0010 | 0.2672±0.0016 | **0.3846±0.0020** | **0.2588±0.0011** |
| | 720 | 0.4383±0.0097 | 0.2913±0.0104 | **0.4241±0.0007** | **0.2816±0.0010** |
| Electricity | 96 | 0.1304±0.0006 | 0.2234±0.0006 | **0.1256±0.0002** | **0.2210±0.0003** |
| | 192 | 0.1482±0.0002 | 0.2403±0.0002 | **0.1451±0.0002** | **0.2397±0.0010** |
| | 336 | 0.1659±0.0006 | 0.2596±0.0006 | **0.1624±0.0010** | **0.2576±0.0009** |
| | 720 | 0.2019±0.0006 | 0.2917±0.0006 | **0.1990±0.0002** | **0.2916±0.0002** |
| ETTh1 | 96 | 0.3752±0.0008 | 0.3999±0.0004 | **0.3700±0.0035** | **0.4001±0.0023** |
| | 192 | **0.4127±0.0012** | **0.4207±0.0006** | 0.4146±0.0012 | 0.4287±0.0013 |
| | 336 | **0.4278±0.0033** | **0.4334±0.0028** | 0.4285±0.0018 | 0.4402±0.0017 |
| | 720 | **0.4462±0.0035** | **0.4637±0.0027** | 0.4670±0.0052 | 0.4768±0.0033 |
| ETTh2 | 96 | **0.2749±0.0005** | **0.3363±0.0006** | 0.2869±0.0039 | 0.3439±0.0016 |
| | 192 | **0.3385±0.0010** | **0.3789±0.0014** | 0.3523±0.0048 | 0.3855±0.0027 |
| | 336 | **0.3288±0.0010** | **0.3823±0.0027** | 0.3779±0.0057 | 0.4112±0.0030 |
| | 720 | **0.3784±0.0010** | **0.4212±0.0009** | 0.3993±0.0054 | 0.4385±0.0038 |
| ETTm1 | 96 | 0.2893±0.0009 | 0.3415±0.0007 | **0.2876±0.0012** | **0.3427±0.0011** |
| | 192 | 0.3316±0.0008 | 0.3695±0.0007 | **0.3296±0.0026** | **0.3688±0.0016** |
| | 336 | 0.3661±0.0022 | 0.3914±0.0012 | **0.3583±0.0015** | **0.3879±0.0016** |
| | 720 | 0.4200±0.0056 | 0.4243±0.0033 | **0.4094±0.0044** | **0.4193±0.0013** |
| ETTm2 | 96 | 0.1647±0.0011 | 0.2538±0.0010 | **0.1637±0.0020** | **0.2537±0.0024** |
| | 192 | 0.2223±0.0018 | 0.2936±0.0014 | **0.2175±0.0011** | **0.2908±0.0013** |
| | 336 | 0.2775±0.0020 | 0.3297±0.0010 | **0.2706±0.0016** | **0.3260±0.0016** |
| | 720 | 0.3648±0.0024 | 0.3833±0.0010 | **0.3539±0.0023** | **0.3799±0.0024** |

Table 14: Multivariate long-term forecasting results with different random seeds in supervised and self-supervised PatchTST/42. The best results are in **bold**.

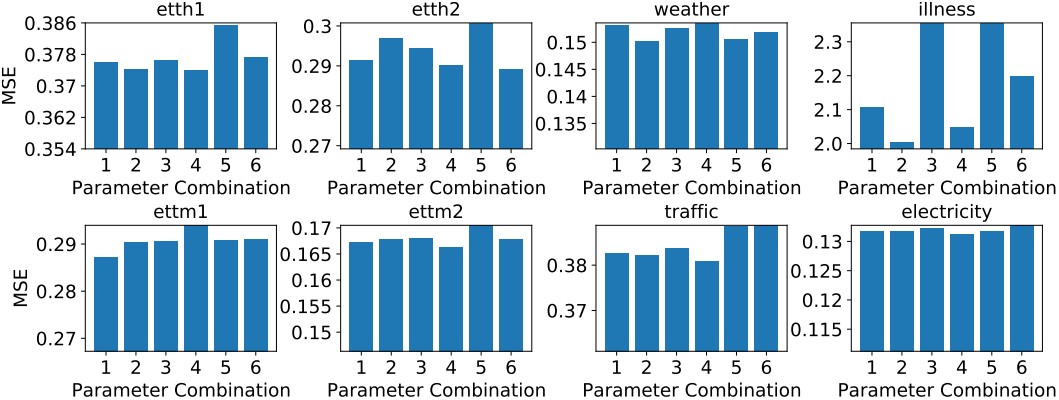

Figure 5: MSE scores with varying model parameters. Each bar indicates the MSE score of a parameter combination. The combinations $(L, D) = (3, 128), (3, 256), (4, 128), (4, 256), (5, 128), (5, 256)$ are orderly labeled from 1 to 6 in the figure. The model is run with supervised PatchTST/42 to forecast 96 steps. For Traffic and Electricity datasets, we reduce the maximum number of epochs to 50 to save computational time.

## A.7 CHANNEL-INDEPENDENCE ANALYSIS

Intuitively, channel-mixing models should outperform the channel-independent ones since they have more flexibility to explore the cross-channel information, while with channel-independence the correlation is indirectly learnt via weight sharing. However, this is contrast to what we have observed. In Section A.7.1 we will provide an in-depth analysis on why channel-independence has better forecasting performance than channel mixing, and in Section A.7.2 we show that channel-independence is a general technique that can be used not only for PatchTST but also for the other models.

### A.7.1 CHANNEL-INDEPENDENCE VS CHANNEL-MIXING

We find 3 key factors that makes channel-independent models more preferable:

- Adaptability: Since each time series is passed through the Transformer separately, it generates its own attention maps. That means different series can learn different attention patterns for their prediction, as shown in Figure 6. In contrast, with the channel mixing approach, all the series share the same attention patterns, which may be harmful if the underlying multivariate time series carries series of different behaviors. Figure 6 reveals an interesting observation that the prediction of unrelated time series relies on different attention patterns while similar series can produce similar maps (e.g. series 11, 25, and 81 contain similar patterns while they are different from others). We suspect this adaptability is one of the main reasons why PatchTST performs much better forecasting than Informer and other channel-mixing models.

- Channel-mixing models may need more training data to match the performance of the channel-independent ones. The flexibility of learning cross-channel correlations could be a double-edged sword, because it may need much more data to learn the information from different channels and different time steps jointly and appropriately, while channel-independent models only focus on learning information along the time axis. We examine this assumption by experiments where we train the models with varying training data size and and the result is shown on left panel of Figure 7. It is clear that channel-independent models converges faster against the size of training data. As what we have observed in the figure and Table 2, the size of those widely used time series datasets may not be large enough for channel-mixing models to obtain similar performances in supervised learning.

- Channel-independent models are less likely to overfit data during training. We record the MSE loss on test data and plot on the right panel of Figure 7. Channel-mixing models show overfitting after a few initial epochs, while channel-independent models continue optimizing the loss with more training epochs. The best trained models are determined by validation data, which are approximately the lowest points in the test loss curves. It is clear that the forecasting performance of channel-independent models are better.

Furthermore, we would like to comment on a few additional technical advantages of channel-independence: **(1)** Possibility of learning spatial correlations across series: Although we haven't focused on this research in our paper, the channel-independence design can be naturally extended to learn cross-channel relationships by using methods like graph neural networks (Cao et al., 2020; Chen et al., 2021). **(2)** Multi-task learning where different loss types can be imposed on different time series where the same underlying Transformer model is shared. **(3)** More robust to noise: If noise is dominant in one or several series, this noise will be projected to other series in the embedding space if we mix channels. Channel independence can mitigate this problem by only retaining the noise in these noisy channels. We can further alleviate the noise by introducing smaller weights to the objective losses that associate with noisy channels.

### A.7.2 PERFORMANCE OF CHANNEL-INDEPENDENCE ON OTHER MODELS

To show that channel-independence is a general technique that can be applied to the other models, we apply it to Informer (Zhou et al., 2021), Autoformer (Wu et al., 2021), and FEDformer (Zhou et al., 2022). The results are shown in Table 15. The channel-independent technique can improve the forecasting performance of those models generally. Although they are still not able to outperform PatchTST which is based on vanilla attention mechanism, we believe that more performance boost and computational reduction can be obtained with more advanced attention designs incorporating the channel-independence architecture.

| Models | | PatchTST/42 | | Informer | | Informer-CI | | Autoformer | | Autoformer-CI | | FEDformer | | FEDformer-CI | |
|---|---|---|---|---|---|---|---|---|---|---|---|---|---|---|---|
| Metric | | MSE | MAE | MSE | MAE | MSE | MAE | MSE | MAE | MSE | MAE | MSE | MAE | MSE | MAE |
| Weather | 96 | 0.152 | 0.199 | 0.300 | 0.384 | **0.174** | **0.232** | 0.266 | 0.336 | **0.227** | **0.289** | 0.217 | 0.296 | **0.214** | **0.278** |
| | 192 | 0.197 | 0.243 | 0.598 | 0.544 | **0.214** | **0.270** | 0.307 | 0.367 | **0.269** | **0.318** | 0.276 | 0.336 | **0.258** | **0.322** |
| | 336 | 0.249 | 0.283 | 0.578 | 0.523 | **0.266** | **0.310** | 0.359 | 0.395 | **0.315** | **0.344** | 0.339 | 0.380 | **0.302** | **0.336** |
| | 720 | 0.320 | 0.335 | 1.059 | 0.741 | **0.327** | **0.356** | 0.419 | 0.428 | **0.384** | **0.389** | 0.403 | 0.428 | **0.374** | **0.369** |
| Traffic | 96 | 0.367 | 0.251 | 0.719 | **0.391** | **0.705** | 0.402 | 0.613 | 0.388 | - | - | 0.587 | 0.366 | - | - |
| | 192 | 0.385 | 0.259 | **0.696** | **0.379** | 0.720 | 0.407 | 0.616 | 0.382 | - | - | 0.604 | 0.373 | - | - |
| | 336 | 0.398 | 0.265 | 0.777 | **0.420** | **0.750** | 0.421 | 0.622 | 0.337 | - | - | 0.621 | 0.383 | - | - |
| | 720 | 0.434 | 0.287 | 0.864 | 0.472 | - | - | 0.660 | 0.408 | - | - | 0.626 | 0.382 | - | - |
| Electricity | 96 | 0.130 | 0.222 | 0.274 | 0.368 | **0.203** | **0.299** | 0.201 | 0.317 | - | - | 0.193 | 0.308 | - | - |
| | 192 | 0.148 | 0.240 | 0.296 | 0.386 | **0.221** | **0.316** | 0.222 | 0.334 | - | - | 0.201 | 0.315 | - | - |
| | 336 | 0.167 | 0.261 | 0.300 | 0.394 | **0.241** | **0.337** | 0.231 | 0.338 | - | - | 0.214 | 0.329 | - | - |
| | 720 | 0.202 | 0.291 | 0.373 | 0.439 | **0.314** | **0.391** | 0.254 | 0.361 | - | - | 0.246 | 0.355 | - | - |
| ILI | 24 | 1.522 | 0.814 | 5.764 | 1.677 | **5.514** | **1.629** | 3.483 | **1.287** | 4.210 | 1.500 | **3.228** | **1.260** | 3.280 | 1.264 |
| | 36 | 1.430 | 0.834 | **4.755** | **1.467** | 5.515 | 1.628 | 3.103 | **1.148** | 2.809 | 1.162 | **2.679** | **1.080** | 2.862 | 1.126 |
| | 48 | 1.673 | 0.854 | **4.763** | **1.469** | 5.263 | 1.574 | 2.669 | **1.085** | 3.218 | 1.267 | **2.622** | **1.078** | 2.834 | 1.150 |
| | 60 | 1.529 | 0.862 | **5.264** | **1.564** | 5.330 | 1.602 | **2.770** | **1.125** | 3.627 | 1.396 | **2.857** | **1.157** | 3.115 | 1.240 |
| ETTh1 | 96 | 0.375 | 0.399 | 0.865 | 0.713 | **0.590** | **0.517** | 0.449 | 0.459 | **0.414** | **0.421** | **0.376** | 0.419 | 0.387 | **0.407** |
| | 192 | 0.414 | 0.421 | 1.008 | 0.792 | **0.677** | **0.566** | 0.500 | 0.482 | **0.453** | **0.448** | **0.420** | 0.448 | 0.439 | **0.438** |
| | 336 | 0.431 | 0.436 | 1.107 | 0.809 | **0.710** | **0.600** | 0.521 | 0.496 | **0.496** | **0.468** | **0.459** | **0.465** | 0.479 | 0.455 |
| | 720 | 0.449 | 0.466 | 1.181 | 0.865 | **0.777** | **0.660** | **0.514** | **0.512** | 0.662 | 0.568 | 0.506 | 0.507 | **0.485** | **0.478** |
| ETTh2 | 96 | 0.274 | 0.336 | 3.755 | 1.525 | **0.390** | **0.410** | 0.358 | 0.397 | **0.337** | **0.373** | 0.346 | 0.388 | **0.297** | **0.348** |
| | 192 | 0.339 | 0.379 | 5.602 | 1.931 | **0.456** | **0.463** | 0.456 | 0.452 | **0.409** | **0.419** | 0.429 | 0.439 | **0.382** | **0.399** |
| | 336 | 0.331 | 0.380 | 4.721 | 1.835 | **0.523** | **0.503** | 0.482 | 0.486 | **0.432** | **0.443** | 0.496 | 0.487 | **0.410** | **0.428** |
| | 720 | 0.379 | 0.422 | 3.647 | 1.625 | **0.843** | **0.661** | 0.515 | 0.511 | **0.443** | **0.463** | 0.463 | 0.474 | **0.422** | **0.444** |
| ETTm1 | 96 | 0.290 | 0.342 | 0.672 | 0.571 | **0.383** | **0.414** | 0.505 | 0.475 | **0.455** | **0.441** | **0.379** | 0.419 | 0.408 | **0.413** |
| | 192 | 0.332 | 0.369 | 0.795 | 0.669 | **0.420** | **0.434** | **0.553** | **0.496** | 0.598 | 0.512 | **0.426** | 0.441 | 0.445 | **0.432** |
| | 336 | 0.366 | 0.392 | 1.212 | 0.871 | **0.465** | **0.467** | 0.621 | 0.537 | **0.566** | **0.504** | **0.445** | 0.459 | 0.476 | **0.452** |
| | 720 | 0.420 | 0.424 | 1.166 | 0.823 | **0.529** | **0.502** | **0.671** | 0.561 | 0.680 | **0.557** | 0.543 | 0.490 | **0.533** | **0.481** |
| ETTm2 | 96 | 0.165 | 0.255 | 0.365 | 0.453 | **0.208** | **0.298** | 0.255 | 0.339 | **0.218** | **0.308** | 0.203 | 0.287 | **0.198** | **0.284** |
| | 192 | 0.220 | 0.292 | 0.533 | 0.563 | **0.274** | **0.345** | **0.281** | 0.340 | 0.281 | **0.339** | 0.269 | 0.328 | **0.259** | **0.320** |
| | 336 | 0.278 | 0.329 | 1.363 | 0.887 | **0.351** | **0.394** | 0.339 | 0.372 | **0.336** | **0.370** | 0.325 | 0.366 | **0.315** | **0.353** |
| | 720 | 0.367 | 0.385 | 3.379 | 1.338 | **0.482** | **0.474** | 0.433 | 0.432 | **0.428** | **0.418** | 0.421 | 0.415 | **0.412** | **0.406** |

Table 15: Channel-independence for other models. CI denote channel-independence. Baselines without CI are cited from Zeng et al. (2022). The better results between CI and non-CI versions are in **bold**. PatchTST/42 is placed on the left for easy reference to other CI-based models. '-' denotes running out of GPU memory even with batch size 1, or exceeding the maximum running time (12 hours).

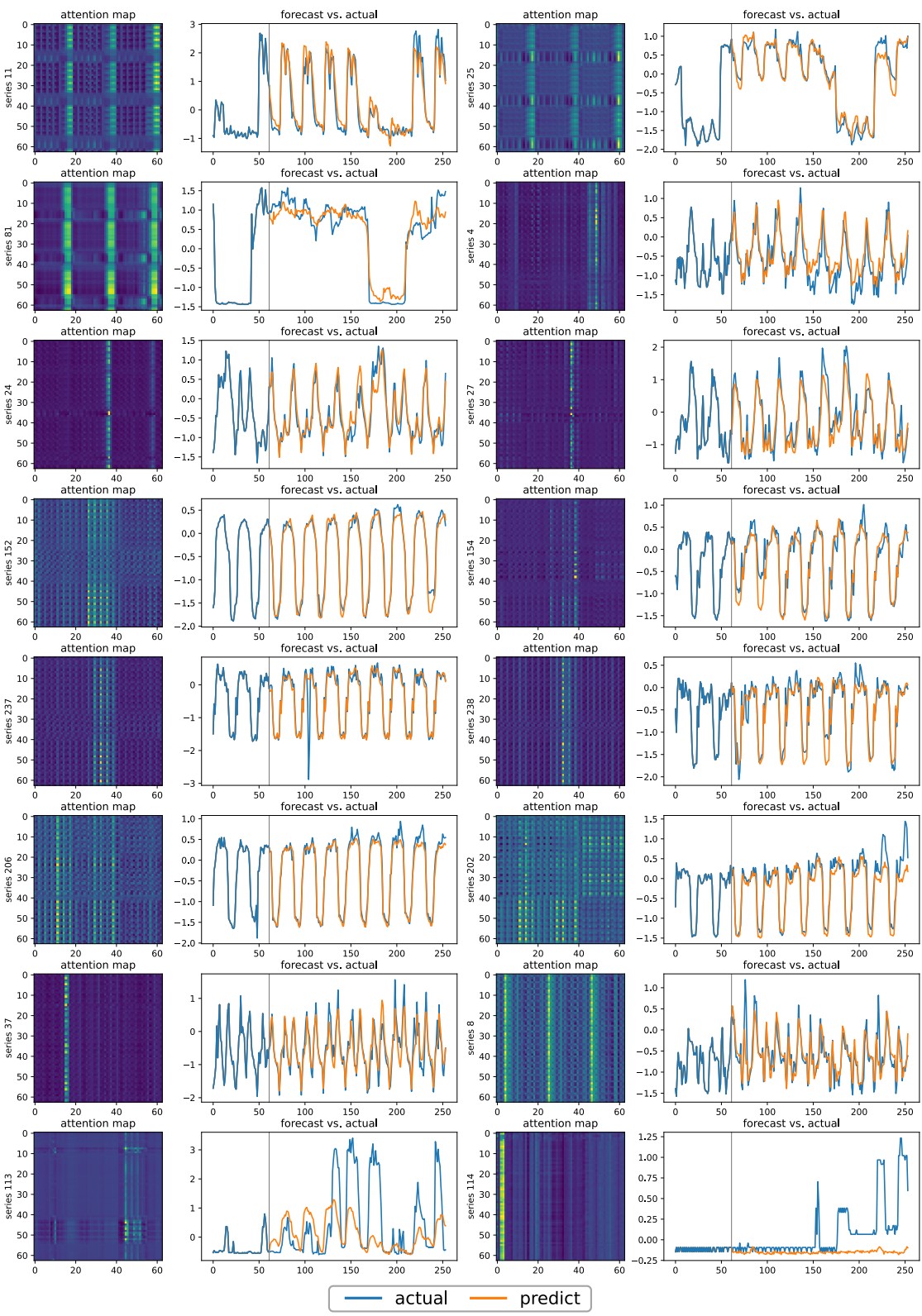

Figure 6: Attention maps and the forecasting of a few time series from Electricity dataset run with supervised PatchTST/64. Attention map is calculated by averaging the attention matrices over all the heads and across all the layers. For each time series, we show the attention map and the prediction in orange. The blue curves are the actual data. The curves before the back lines are the actual input data. Channel-independence design allow each series to learn its own attention map for forecasting in which the pattern can be more similar for more correlated series and different otherwise.

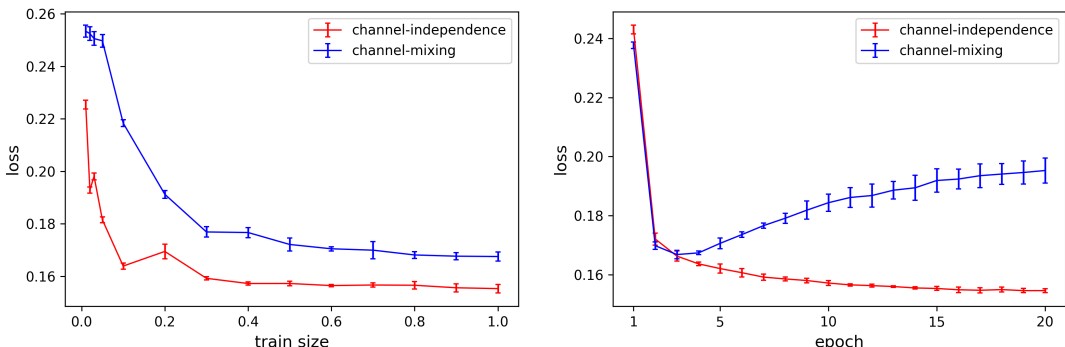

Figure 7: Channel-independence vs channel-mixing on Weather dataset. The base model is PatchTST/42, and the prediction length is 96. We plot the mean values and error bars with 5 different random seeds: {2019, 2020, 2021, 2022, 2023}. **Left Panel:** Test loss vs train size. Here, train size denotes the fraction of the training data that is used to learn the model from scratch. Channel-independence contributes to a quicker convergence as more training data is available. **Right Panel:** Test loss vs epochs. Here, we use full train data and plot the first 20 epochs. Channel-mixing model quickly overfits the data.

