# OpenReview forum: "A Time Series is Worth 64 Words:  Long-term Forecasting with Transformers"
_ICLR.cc/2023/Conference — ICLR 2023 poster_

### Official Review · Reviewer_3Fzy · 2022-10-23

**Confidence:** 4
**Correctness:** 3
**Technical Novelty And Significance:** 2
**Empirical Novelty And Significance:** 2
**Recommendation:** 5

**Clarity, Quality, Novelty And Reproducibility:**

The paper is well-orgonized and clarified. However, its novelty is somewhat weak.

**Strength And Weaknesses:**

Strength: Multivariate time series forecasting and self-supervised representation learning are important tasks for time series. The proposed methods have some improvements on both tasks.

Weaknesses: I have the following questions.
(i) Segmentation of time series is a naive extension from computer vision (MAE). Thus, it lacks novelty in my perspective.
(ii) In my view, channel-independence is an orthogonal technic compared with other methods. In other words, the authors can perform channel-independence on other baselines. Can the authors show the performances with channel-independence in Table 3 and Table 4?
(iii) Can you provide some analysis for channel-independence? Since different channels have correlations compared with others, why channel-indpendence can improve performance?

**Summary Of The Paper:**

The authors propose an efficient design of Transformer-based models for multivariate time series forecasting and self-supervised representation learning. There are two main components: (i) segmentation of time series into subseries-level patches which are served as input tokens to Transformer; (ii) channel-independence where each channel contains a single univariate time series that shares the same embedding and Transformer weights across all the series. The experiment results show the effectiveness of their method.

**Summary Of The Review:**

The paper proposes two main technics for time series tasks. Even though the experiment results are better than baselines, the improvement is marginal and there miss some deep analysis. Thus, I vote for marginally below the acceptance threshold.

---

> ### Author Response · Authors · 2022-11-19
> **Response to Reviewer 3Fzy (Q1, Q2)**
>
> We appreciate the reviewer’s comments. We update our paper, code link and address your concerns here:
>
> > Q1: Segmentation of time series is a naive extension from computer vision (MAE). Thus, it lacks novelty in my perspective.
>
> A1: Thank you for the comment. We would like to claim our novelty in the following points:
>
> 1. Although segmentation is a general technique that has been considered in various scenarios, **effectively combining it with channel-independence and applying to time series Transformer to obtain state-of-the-art performance is non-trivial** and has not been studied before. In fact, we will not get SOTA performance if each of the techniques stands alone as we demonstrated in Table 10. Either the mixed-channels (column P), or the local information not being captured (column CI) can be factors for the performance degradation. Moreover, the computational cost can be significantly higher for large datasets as shown in Table 1 if patching is not enabled.
>
> 2. **Patching on time series is also different from previous methods in terms of capturing locality.** Over the last couple of years, most of the time series models that we mentioned in Section 2 focus on designing novel mechanisms to reduce the complexity of original attention mechanisms, and most of them are either using point-wise attention (like Informer) which ignores locality, or replacing the vanilla attention with hand-crafted modules that can extract some specific series-wise information (like Autoformer, FEDformer). Instead, we show that simply using patching to capture locality with the vanilla attention can maintain all the information from raw signal, and obtain significant improvements compared to the previous Transformer-based models. We believe that our approach can lead to several research developments in this direction.
>
> 3. **We provide an effective way of applying patching to multivariate time series benefitting from channel-independence.** The patch design can naturally incorporate with channel-independent architecture. Patching will create an additional dimension to the input data (input data from dimension $M \times L$ to $M \times P \times N$ where $L$ is the sequence length, $M$ the number of channels, $P$ the patch size, and $N$ the number of patches). To deal with it, either this extra dimension needs to be stacked over all the channels $(M \cdot P) \times N$ or the Transformer has to be re-designed to take a new input. With a channel-independent structure, however, the patched input can naturally be fed to the vanilla Transformer without any complex modification. Results in Table 10 demonstrates the effectiveness of this combination.
>
> > Q2: In my view, channel-independence is an orthogonal technic compared with other methods. In other words, the authors can perform channel-independence on other baselines. Can the authors show the performances with channel-independence in Table 3 and Table 4?
>
> A2: We agree with your comment that channel-independence can be applied to any Transformer architecture. We apply channel-independence to other baselines such as Informer, Autoformer, and FEDformer, and show the results in the updated pdf (**Table 15**). In general, channel-independence can improve the forecasting performance on those models. Although they are not able to outperform PatchTST, which is based on vanilla attention mechanism, we believe that more performance boost and computational reduction can be possible with more advanced attention designs to incorporate the channel-independence architecture. We don’t run channel-independence versions for self-supervised learning as PatchTST in Table 4 since those baseline models are mainly designed for supervised learning.

---

> ### Author Response · Authors · 2022-11-19
> **Response to Reviewer 3Fzy (Q3, Q4)**
>
> > Q3: Can you provide some analysis for channel-independence? Since different channels have correlations compared with others, why channel-indpendence can improve performance?
>
> Thank you for this very thoughtful question. We add a new section in **appendix (Section A.7)** to provide an in-depth analysis of why channel-independence can improve the performance and point out a few technical advantages. We summarize the key points here due to the word limit:
>
> Key reasons:
>
> 1. Capability of getting channel-specific attention maps.
>
> 2. Channel-mixing models may need more training data to match the performance of the channel-independent ones.
>
> 3. Channel-independent models seem less likely to overfit data during training.
>
> Additional technical advantages:
>
> 1. Possibility of learning spatial correlations across series.
>
> 2. Convenience on multi-task learning.
>
> 3. More robust to noise.
>
> > Q4: Even though the experiment results are better than baselines, the improvement is marginal
>
> A4: Thanks for bringing up this concern. We probably have different thoughts on that and we would like to address it in the following points:
>
> 1. Compared to the current Transformer-based models (X-former), the improvement of our model is significant. Taking large Traffic data with 96 steps forecasting for example, the improvement to the best FEDformer model can be as large as 37.5% and to the famous Informer, the improvement is even more significant.
>
> 2. When compared with all the other non-Transformer-based models, especially the recent DLinear model as a strong baseline, we are still the SOTA.
>
> 3. Besides the promising numerical results, we think our approach provides another methodology for efficiently applying Transformer to time series, since the framework is easy to implement and has good potential for improvement.

---

### Official Review · Reviewer_2c5h · 2022-10-24

**Confidence:** 4
**Correctness:** 4
**Technical Novelty And Significance:** 3
**Empirical Novelty And Significance:** 3
**Recommendation:** 8

**Clarity, Quality, Novelty And Reproducibility:**

This paper is clearly motivated and well-written. The idea of utilizing patches to deal with time series is interesting although channel independence is hardly novel given that previous work have explored this setting in time series forecasting. The paper provides some details of its implementation, however, it may need some efforts to reproduce its results as there is no sufficient details, e.g. hyperparameters and code, for reproduction.

**Strength And Weaknesses:**

Strength

*  The idea of utilizing patch like ViT to reduce sequence length and better capture locality is quite simple and effective
* Extensive experiments on multiple datasets show that it can outperform recently state-of-the-art methods in different forecasting horizons and ablation study regarding patches and channel independence is conducted across multiple datasets.
*  This paper is well-organized and easy to understand

Weaknesses

* Channel independence is hardly novel as previous work, e.g. DeepAR [1] and LogTrans, also have this assumption. These two work model multivariate time series by sharing backbone parameters (LSTM/Transformer) of different channels during training, which can reduce parameter numbers and overfitting issue.

* In experimental setups, "For Transformer-based models, the default lookback window is L = 96". However, this look-back window length can be quite short and may not be ideal especially when forecasting window is long, e.g. 336. In figure 2 (Electricity-T=720), best look-back window for Autoformer is  192/336 rather than 96. This setting make me have some concerns that baselines may be under-estimated.

* Self-supervised methods are only a little better and standard supervised fine-tuning. I am wondering if authors can have multiple runs and report mean and standard deviation so that readers may know how significant it is.

[1] Flunkert, Valentin et al. “DeepAR: Probabilistic Forecasting with Autoregressive Recurrent Networks.” ArXiv abs/1704.04110 (2020): n. pag.



**Summary Of The Paper:**

This paper proposes an efficient design of Transformer-based models (PatchTST) for multivariate time series forecasting and self-supervised representation learning. It segments time series into patches following similar philosophy of ViT and assume channels are independent. Extensive experiments on 8 eight datasets show that it can outperform state-of-the-art models while requires less compute.

**Summary Of The Review:**



 I update my score to 8 since the authors' response resolve concerns and provide source code on Github.

\------------------------------------------------------------------------------

The idea of utilizing patch to deal with time series is interesting and have good empirical results. However, channel independence is hardly novel. In addition, I have concerns regarding experimental settings and am not sure if baselines are under-estimated and results statistically significant.

---

> ### Author Response · Authors · 2022-11-19
> **Response to Reviewer 2c5h (Q1)**
>
> We appreciate the reviewer’s comments. We update our paper, code link and address your concerns below:
>
> > Q1: Channel independence is hardly novel as previous work, e.g. DeepAR [1] and LogTrans, also have this assumption. These two work model multivariate time series by sharing backbone parameters (LSTM/Transformer) of different channels during training, which can reduce parameter numbers and overfitting issue.
>
> A1: We thank the reviewer for bringing up that point. We claim our novelty in the following points:
>
> 1. **Channel-independence is fundamentally different from LogTrans and DeepAR design**, although our method shares some similarities with theirs such as sharing model weights across different channels. In the later approaches, each input batch to the LSTM/Transformer contains samples that are extracted randomly from **different channels** and **at random time steps**, while in our method, batch samples are drawn from **the same time step across all channels**. The former sampling assumes samples across different channels and time steps have similar underlying dynamics, while we relax that assumption and only require samples to have similar dynamics across time. Since samples from the same time step across channels do not necessarily have the same dynamics, we can further design models to learn the correlation between channels which we will study in future work.
>
> 2. **Effectively combining channel-independence with patching and applying to time series Transformer to obtain state-of-the-art performance is non-trivial** and has not been studied before. In fact, we will not get SOTA performance if each of the techniques stands alone as we demonstrated in Table 10. Either the mixed-channels (column P), or the local information not being captured (column CI) can be factors for the performance degradation. Moreover, the computational cost can be significantly higher for large datasets as shown in Table 1 if we don't apply this combination but just use channel-independence.
>
> 3. **Benefitting from channel-independence, we provide an effective way of applying patching to multivariate time series.** The patch design can naturally incorporate with channel-independent architecture. Patching will create an additional dimension to the input data (input data from dimension $M \times L$ to $M \times P \times N$ where $L$ is the sequence length, $M$ the number of channels, $P$ the patch size, and $N$ the number of patches). To deal with it, either this extra dimension needs to be stacked over all the channels $(M \cdot P) \times N$ or the Transformer has to be re-designed to take a new input. With a channel-independent structure, however, the patched input can naturally be fed to the vanilla Transformer without any complex modification. Results in Table 10 demonstrates the effectiveness of this combination.
>
> Additionally, we make an in-depth analysis of why channel-independence improves the performance and point out a few technical advantages of it in **appendix (A.7)** for your reference .

---

> ### Author Response · Authors · 2022-11-19
> **Response to Reviewer 2c5h (Q2, Q3, Q4)**
>
> > Q2: In experimental setups, "For Transformer-based models, the default lookback window is L = 96". However, this look-back window length can be quite short and may not be ideal especially when forecasting window is long, e.g. 336. In figure 2 (Electricity-T=720), best look-back window for Autoformer is 192/336 rather than 96. This setting make me have some concerns that baselines may be under-estimated.
>
> A2: We thank the reviewer for this constructive comment. To address this issue, We run experiments on FEDformer, Autoformer and Informer with 6 different look-back windows $L\in$ { $24,36,48,60,104,144$ } for ILI dataset and $L\in$ { $24,48,96,192,336,720$ } for others, and for each different prediction length on each dataset, we select best one from the six results. This creates a strong baseline for those models. We update the baseline results in all the related tables in our paper. As observed from the updated table, our improvement is still significant compared to the best baselines.
>
> > Q3: Self-supervised methods are only a little better and standard supervised fine-tuning. I am wondering if authors can have multiple runs and report mean and standard deviation so that readers may know how significant it is.
>
> A3: Thank you for your suggestion. We have performed pretraining experiments with multiple seed numbers and provided the results in the updated paper (**Table 14**). We have also done other robustness checks in **appendix (A.6)** for your reference.
>
> > Q4: The paper provides some details of its implementation, however, it may need some efforts to reproduce its results as there is no sufficient details, e.g. hyperparameters and code, for reproduction.
>
> A4: Thanks for asking about the implementation. The codes are available in this repository: https://github.com/PatchTST/PatchTST. All the hyperparameters that we choose as well as other implementation details can be found here. Please note that this is not the final official repo because we want to maintain anonymity. Once the anonymous review ends we will update it with a final official link.

---

### Official Review · Reviewer_zBqz · 2022-10-24

**Confidence:** 4
**Correctness:** 3
**Technical Novelty And Significance:** 2
**Empirical Novelty And Significance:** 2
**Recommendation:** 5

**Clarity, Quality, Novelty And Reproducibility:**

Clarity: limited.

Quality: fair.

Novelty: limited.

Reproducibility: fair

**Strength And Weaknesses:**

Strength
1. The essential idea of the paper is well motivated and presented.
2. Extensive experiments are done to compare the proposed model with a variety of baselines.

Weakness:
1. The proposed patching is not novel as LogTrans (Li et al. 2019) has already proposed to use convolution before transformer to enhance locality. The linear processing in PatchTST with fixed path size and stride is not different from 1D convolution, even though a larger grid of configuration is investigated in the experiments.
2. The idea of masked modeling in representation learning is also marginally novel. The motivation of such a strategy is vague in certain parts. For example, in section 3.2, paragraph 2, _Since time series is often temporally redundant, the masked values at the current time step can be easily referred from neighboring values, which makes the reconstruction process trivial and thus the representation may not carry important abstract information_, the triviality here is not clear: in which case ("redundancy") the current time step can be too "easily" to refer? Masked modeling itself aims to refer the masked token/patch from contextual objects, why is it deemed trivial in time series? Moreover, in paragraph 3, the prediction in multivariate time series can also employ certain degree of weight sharing to avoid $(L\cdot D) \times (M\cdot T)$ parameter matrices.
3. Detailed questions:
+ in supervised learning, the model is said to have a flatten layer with linear head to get univariate time series. When the patches have overlapping, how is the prediction within the overlapping determined?
+ how is the linear probing and end-to-end fine-tuning done after representation learning? Is linear probing simply reconstructing masked patches again but not update the backbone layers except the final linear head?

**Summary Of The Paper:**

This work proposed a transformer architecture that forecast multivariate time series channel-wise. In particular, the model employs patching by grouping values in temporal neighborhoods, so that the locality is enhanced and efficiency is improved. The model can also be trained in a self-supervised way by masking random patches, and it is shown such self-supervised pre-training can lead to superior performance with proper fine-tuning.

**Summary Of The Review:**

Overall I think the current version of this paper does not meet the bar of ICLR due to limited novelty and insufficient clarity.

---

> ### Author Response · Authors · 2022-11-19
> **Response to Reviewer zBqz (Q1)**
>
> We appreciate the reviewer’s comments. We update our paper, code link and address your concerns below:
>
> > Q1: The proposed patching is not novel as LogTrans (Li et al. 2019) has already proposed to use convolution before transformer to enhance locality. The linear processing in PatchTST with fixed path size and stride is not different from 1D convolution, even though a larger grid of configuration is investigated in the experiments.
>
> A1: Thank you for this thoughtful question. We would like to claim our novelty in the following points:
>
> 1. **Patching on time series is different from the convolution method by LogTrans.** We agree that their setting enhances the locality, since the query and key are coming from convolution. However the value is based on a single time step, which still gives a point-wise attention map.  It is analogous to finding the correlations between each single letter in a sentence, which could be less reasonable than finding the correlations between words/subwords. Thus, we think of patching as a more natural way to capture the locality.  Also, the performance of LogTrans is limited based on Table 3. Besides LogTrans, patching is also different from other manually designed modules which extract some specific series-wise information. It is simple to implement while maintaining all the information from the raw signal.  We believe that our approach can lead to several research developments in this direction.
>
> 2. **Effectively combining patching with channel-independence and applying to time series Transformer to obtain state-of-the-art performance is non-trivial** and has not been studied before. In fact, we will not get SOTA performance if each of the techniques stands alone as we demonstrated Table 10. Either the mixed-channels (column P), or the local information not being captured (column CI) can be factors for the performance degradation. What’s more, the computational cost can be significantly higher for large datasets as shown in Table 1 if patching is not enabled.
>
> 3. **We provide an effective way of applying patching to multivariate time series benefitting from channel-independence.** The patch design can naturally incorporate with channel-independent architecture. Patching will create an additional dimension to the input data (input data from dimension $M \times L$ to $M \times P \times N$ where $L$ is the sequence length, $M$ the number of channels, $P$ the patch size, and $N$ the number of patches). To deal with it, either this extra dimension needs to be stacked over all the channels $(M \cdot P) \times N$ or the Transformer has to be re-designed to take a new input. With a channel-independent structure, however, the patched input can naturally be fed to the vanilla Transformer without any complex modification. Results in Table 10 demonstrates the effectiveness of this combination.

---

> ### Author Response · Authors · 2022-11-19
> **Response to Reviewer zBqz (Q2)**
>
> > Q2: The idea of masked modeling in representation learning is also marginally novel. The motivation of such a strategy is vague in certain parts. For example, in section 3.2, paragraph 2, Since time series is often temporally redundant, the masked values at the current time step can be easily referred from neighboring values, which makes the reconstruction process trivial and thus the representation may not carry important abstract information, the triviality here is not clear: in which case ("redundancy") the current time step can be too "easily" to refer? Masked modeling itself aims to refer the masked token/patch from contextual objects, why is it deemed trivial in time series? Moreover, in paragraph 3, the prediction in multivariate time series can also employ certain degree of weight sharing to avoid  $(L \cdot D) \times (M \cdot T)$ parameter matrices.
>
> A2: We thank the reviewer for the detailed question. We modify the corresponding explanations in the paper based on your comment to make it clear. What we mean “temporally redundant” here is that the time series often contains several smooth regions. In the smooth regions where the time value x(t) at time t is not very different from values in the previous time (t-1) and future time (t+1), the model can learn to copy the values from either x(t-1) or x(t+1) to replace the masked value x(t) instead of the learning a more global abstract representation. Therefore, if the masking is performed randomly over time, it is possible that the pre-trained solution may not be useful. This point has been made from the paper [1] and it is the reason the authors in the paper propose different masking techniques to avoid trivial solutions. This observation is also true in computer vision where masking at pixel level may not produce a good representation.
>
> By masking at patch level, we remove the entire contiguous time windows. The model needs to capture meaningful global patterns in order to reconstruct the windows accurately.
>
> In the paper [1], the representation variables contain $D$ dimensions and $L$ time steps. In order to perform forecasting $T$ time steps ahead on $M$ time series, there requires a map from the representation to the forecasting space. As shown in Table 8 of the paper [2], the number of parameters of Pyraformer (a model that only uses the Transformer decoder like in [1]) is increased significantly due to the size of the last linear layer, which is the same as our observation.
>
> We agree with you that there may exist a certain degree of weight sharing to reduce the size of the linear mapping. However, this will likely introduce certain model bias and can degrade the prediction performance. To better address this concern, we design another type of head (denoted as Head 1) that first maps hidden dimension $D$ to feature dimension $M$, then maps context window $L$ to prediction length $T$. Compared to the head (denoted as Head 2) that directly maps $(L⋅D)$ to $(M⋅T)$, the parameter number of Head 1 is reduced from $(L⋅D⋅M⋅T)$ to $(M⋅D+L⋅T)$. We apply it on the ETTm1 dataset with the same model as in Table 7, column Original. Unfortunately, we see that this parameter reduction will hurt the accuracy as shown in the table below.
>
> |   |  Head 1  | | Head 2|  |
> |---|---|---|---|---|
> | $T$ |MSE|MAE|MSE|MAE|
> |96|0.373|0.409|**0.324**|**0.370**|
> |192|0.403|0.428|**0.373**|**0.398**|
> |336|0.443|0.451|**0.415**|**0.421**|
> |720|0.489|0.483|**0.480**|**0.459**|
>
> Despite these considerations, masking itself is not the main novelty that we are trying to propose in the paper, since it is a general technique that was widely used in representation learning.  Instead, we try to claim the feasibility of our model on self-supervised learning tasks, since it already beats the other cutting-edge self-supervised models by a large margin (Table 6), as well as shows promising potential to outperform the supervised time series models (Table 4). We believe that more performance boost can be observed with more sophisticated design on self-supervised learning tasks, like the Swin Transformer [3] in computer vision, which is also an important part of our future work.
>
> References:
>
> [1] George Zerveas, Srideepika Jayaraman, Dhaval Patel, Anuradha Bhamidipaty, and Carsten Eickhoff.
> A transformer-based framework for multivariate time series representation learning. In Proceedings of the 27th ACM SIGKDD Conference on Knowledge Discovery & Data Mining, pp. 2114–2124 (2021).
>
> [2] Ailing Zeng, Muxi Chen, Lei Zhang, and Qiang Xu. Are transformers effective for time series forecasting? arXiv:2205.13504 (2022).
>
> [3] Ze Liu, Yutong Lin, Yue Cao, Han Hu, Yixuan Wei, Zheng Zhang, Stephen Lin and Baining Guo. Swin Transformer: Hierarchical Vision Transformer Using Shifted Windows. In Proceedings of the IEEE/CVF International Conference on Computer Vision (ICCV), pp. 10012-10022 (2021).

---

> ### Author Response · Authors · 2022-11-19
> **Response to Reviewer zBqz (Q3, Q4)**
>
> > Q3: In supervised learning, the model is said to have a flatten layer with linear head to get univariate time series. When the patches have overlapping, how is the prediction within the overlapping determined?
>
> A3: We apologize for the confusion. As demonstrated in Figure 1, patching is a pre-processing step of data to help the model better understand input series, which is analogous to partitioning a sentence into words to understand it. We don’t apply patch operator on the forecast sequence. In the forecasting tasks, each time we will receive an input data with a look-back window $L$, then the $L$-length sequence will be split into some patches (either overlapped or not), and all those patches are used together to predict the future $T$ steps. The whole $T$-length forecast series is generated at once by the model.
>
> The role of a flatten layer with a linear head is to map the representation $z^{(i)}\in \mathbb{R}^{D\times N}$ to the output series $\hat{x}^{(i)}\in\mathbb{R}^{1\times T}$, as illustrated on the left side of Figure 1 (b). Basically first a flatten layer transforms the representation $z^{(i)}$ of dimension $D \times N$ into a flattened vector of dimension $D·N$, then a linear layer maps this vector to the output dimension $T$.
>
> > Q4: How is the linear probing and end-to-end fine-tuning done after representation learning? Is linear probing simply reconstructing masked patches again but not update the backbone layers except the final linear head?
>
> A4: We are sorry for not being clear. Linear probing and fine-tuning are applied to downstream tasks where the task is not to reconstruct masks. Instead, the task here is forecasting.  Linear probing only updates the last linear layer while the backbone layers are frozen. In contrast, fine-tuning updates the entire model during training. We here provide a good reference [4] to illustrate those two methods.  And for fine-tuning we are also using an optimized two step strategy as proposed in [4].
>
> References:
>
> [4] Ananya Kumar, Aditi Raghunathan, Robbie Matthew Jones, Tengyu Ma, and Percy Liang. Fine-tuning can distort pretrained features and underperform out-of-distribution. In International Conference on Learning Representations (2022).

---

### Official Review · Reviewer_FEPh · 2022-10-25

**Confidence:** 3
**Correctness:** 3
**Technical Novelty And Significance:** 2
**Empirical Novelty And Significance:** 3
**Recommendation:** 6

**Clarity, Quality, Novelty And Reproducibility:**


Clarity: The paper is well written and I enjoy reading it.

Quality: The overall quality is good. After the introduction section, the authors first discuss the model settings and then switch to several experiments to illustrate the efficiency of the proposed model.

Novelty: Despite the strong numerical performance, the technical novelty is kind of limited as the proposed model structure is mainly borrowed from the standard VIT.

Reproducibility: The paper doesn't include the codes/git. The model parameters are reported but some other parameters, such as learning rate, and optimizer are not reported. At the current stage, I haven't checked the reproducibility.

**Strength And Weaknesses:**

Strength:
1. Good numerical performance forecasting, reaching the SOTA results on several real datasets.

Weakness:
1. Based on the current presentation, the proposed model is analogous to the standard vision transformer (VIT). The technical novelty is limited.
2. The pretraining datasets are relatively small compared to datasets in CV/NLP fields and the domains of the datasets merely share the mutual underlying knowledge. Combining with marignal improvements reported in Table 4 and Table 5, I tends to believe that the pretraining stage isn't meaningful to the considered datasets.



**Summary Of The Paper:**

This paper studies an efficient way to use the transformer-based model in time-series forecasting tasks. The proposed model, PatchTST, first folds the sequence into several patches which significantly reduces the total sequence length and then splits the multi-channel forecasting signal into independent tasks. In numerical tests on several real datasets, the proposed model reaches the SOTA performance. The authors also test their model on self-supervised learning and verify a promising sign that the transformer-based models could be fit on time-series pretraining tasks.

**Summary Of The Review:**

This paper proposes an efficient transformer-based model in time-series forecasting tasks and strong numerical performance is presented. My major concern is the technical novelty as the proposed model is mainly a direct application of standard VIT.


Minor issues:

1. The "64 words" in the title is a little bit misleading. In the famous VIT paper "An Image is Worth 16x16 Words: Transformers for Image Recognition at Scale", the patch size is 16 x 16. However, the patch size/length in the current paper is around 8-16 which doesn't match 64. I suggest authors modify the title.

2. More ablation studies. In the current presentation, the ablation without instance norm is not included. In the literature, using the instance norm or the equivalent RevIN in [1] could significantly improve the numerical performance. In order to further highlight the effectiveness of the patch and the channel independence ideas, I'm wondering if adding one more ablation study related to the instance norm will be helpful.

3. Robustness test. In time-series forecasting tasks, usually, the results are very sensitive to the model/training parameters.  How about the proposed model? Can we also include a section to discuss the robustness of the proposed model?


[1] Kim, T., Kim, J., Tae, Y., Park, C., Choi, J. H., & Choo, J. (2021, September). Reversible instance normalization for accurate time-series forecasting against distribution shift. In International Conference on Learning Representations.

---

> ### Author Response · Authors · 2022-11-19
> **Response to Reviewer FEPh (Q1, Q2, Q3)**
>
> We appreciate the reviewer’s comments. We update our paper, code link, and address your concerns here:
>
> > Q1: Based on the current presentation, the proposed model is analogous to the standard vision transformer (VIT). The technical novelty is limited.
>
> A1: Thank you for the comment. We would like to claim our novelty in the following points:
>
> 1. Although patching is a general technique that has been considered in various scenarios, **effectively combining it with channel-independence and applying to time series Transformer to obtain state-of-the-art performance is non-trivial** and has not been studied before. In fact, we will not get SOTA performance if each of the techniques stands alone as we demonstrated in Table 10. Either the mixed-channels (column P), or the local information not being captured (column CI) can be factors for the performance degradation. Moreover, the computational cost can be significantly higher for large datasets as shown in Table 1 if patching is not enabled.
>
> 2. **Patching on time series is also different from previous methods in terms of capturing locality.** Over the last couple of years, most of the time series models that we mentioned in Section 2 focus on designing novel mechanisms to reduce the complexity of original attention mechanisms, and most of them are either using point-wise attention (like Informer) which ignores locality, or replacing the vanilla attention with hand-crafted modules that can extract some specific series-wise information (like Autoformer, FEDformer). Instead, we show that simply using patching to capture locality with the vanilla attention can maintain all the information from raw signal, and obtain significant improvements compared to the previous Transformer-based models. We believe that our approach can lead to several research developments in this direction.
>
> 3. **We provide an effective way of applying patching to multivariate time series benefitting from channel-independence.** The patch design can naturally incorporate with channel-independent architecture. Patching will create an additional dimension to the input data (input data from dimension $M \times L$ to $M \times P \times N$ where $L$ is the sequence length, $M$ the number of channels, $P$ the patch size, and $N$ the number of patches). To deal with it, either this extra dimension needs to be stacked over all the channels $(M \cdot P) \times N$ or the Transformer has to be re-designed to take a new input. With a channel-independent structure, however, the patched input can naturally be fed to the vanilla Transformer without any complex modification. Results in Table 10 demonstrates the effectiveness of this combination.
>
> > Q2: The paper doesn't include the codes/git.
>
> A2: Thanks for asking about the codes. The codes are available in this repository: https://github.com/PatchTST/PatchTST. This is not the final official repo because we want to maintain anonymity. Once the anonymous review ends we will update it with a final official link.
>
> > Q3: The pretraining datasets are relatively small compared to datasets in CV/NLP fields and the domains of the datasets merely share the mutual underlying knowledge. Combining with marignal improvements reported in Table 4 and Table 5, I tends to believe that the pretraining stage isn't meaningful to the considered datasets.
>
> A3: We appreciate you for this thoughtful question. The foundation models in the time series field are still in a very early stage, and here we try to claim the feasibility of our model on those self-supervised learning tasks. Although the pretraining is not comparable with the scale in CV/NLP, our model has already beat the other cutting-edge self-supervised time series models by a large margin (Table 6), as well as shown promising potential to outperform the supervised time series models (Table 4). We believe that more performance boost can be observed in future research, either with more emerging standard large time series datasets like ImageNet in CV, or with more sophisticated design of self-supervised learning tasks.

---

> ### Author Response · Authors · 2022-11-19
> **Response to Reviewer FEPh (Q4, Q5, Q6)**
>
> > Q4: The "64 words" in the title is a little bit misleading. In the famous VIT paper "An Image is Worth 16x16 Words: Transformers for Image Recognition at Scale", the patch size is 16 x 16. However, the patch size/length in the current paper is around 8-16 which doesn't match 64. I suggest authors modify the title.
>
> A4: We apologize for the confusion. The 64 words in our title is based on the number of patches that are fed to the Transformer. Each patch contains a certain semantic meaning similar to the word in the sentence. Surely, the number of patches can vary based on the applications, so the number 64 is merely a wordplay.
>
> From the ViT paper, since each figure contains 224x224 pixels, the “16x16 words” corresponds to a patch size 14x14, which is the model ViT-H/14. We also cite the official reply from the ViT authors in openreview about their title:
>
> “This is merely a wordplay based on the fact that our largest model (H/14), when trained on the standard ImageNet resolution 224x224 pixels, splits the input image into 16x16=256 patches, and we feed these patches to a transformer in the same way words are fed to transformers in NLP.”
>
> > Q5: More ablation studies. In the current presentation, the ablation without instance norm is not included. In the literature, using the instance norm or the equivalent RevIN in [1] could significantly improve the numerical performance. In order to further highlight the effectiveness of the patch and the channel independence ideas, I'm wondering if adding one more ablation study related to the instance norm will be helpful.
>
> A5: Thank you for the insightful suggestion. We do the ablation study on instance norm and report the results in the updated pdf (**Table 11**). Although the models perform slightly better with instance normalization, compared to other Transformer-based models, the proposed approach still achieves significantly better forecasting even without instance normalization. This is to highlight that the main source of improvement comes from patching and channel-independence designs.
>
> > Q6: Robustness test. In time-series forecasting tasks, usually, the results are very sensitive to the model/training parameters. How about the proposed model? Can we also include a section to discuss the robustness of the proposed model?
>
> A6: Thank you for your suggestion. To better address this concern, we add a section in **appendix (A.6)** for robustness analysis. We run our PatchTST with varying hyper-parameter settings, in particular the number of layers and model dimensions, to see if there are large performance gaps with different model parameters. Figure 5 demonstrates that across all the datasets, the variance in MSE score is insignificant, except illness data where the number of samples are very small. We train all the models with the same training parameters: we use Adam algorithm with the learning rate defined by the learning rate finder technique (similar to this https://pytorch.org/ignite/generated/ignite.handlers.lr_finder.FastaiLRFinder.html) and with super convergence learning rate scheduler (https://arxiv.org/pdf/1708.07120.pdf).
>
> We also run experiments with different seed numbers for both supervised training and self-supervised training to report the variances in Table 14, which again shows negligible variance across different runs.

---

### Author Response · Authors · 2022-11-19
**Summary of Revisions**

**We appreciate all the reviewers for their time to provide valuable feedback and suggestions to improve our paper substantially.**

**We find the positive feedback for our paper as follows:**

1. This paper is well-organized and easy to understand. The essential idea of the paper is well motivated and presented. Multivariate time series forecasting and self-supervised representation learning are important tasks for time series.

2. The idea of utilizing patch to reduce sequence length and better capture locality is quite simple and effective.

3. Extensive experiments on multiple datasets show that it can outperform recently state-of-the-art methods in different forecasting horizons and ablation study regarding patches and channel independence is conducted across multiple datasets.

**We make the following key updates to the paper to better address the reviewers’ concerns:**

1. We add a new section in appendix (Section A.7) to provide an in-depth analysis of why **channel-independence** can improve the performance and point out a few technical advantages of it.

2. We add a section in appendix (Section A.6) for **robustness** analysis, by running experiments with different random seeds and model hyper-parameters.

3. We add an ablation study on **instance normalization** in Appendix (Section A.4.4).

4. We create **strong baselines** for FEDformer, Autoformer and Informer to avoid possible under-estimation. We run these models with six different look-back windows, and choose the best results. The baseline results are updated. Our improvement is still significant and our arguments are not affected.

5. We modified some expressions in the draft based on reviewers’ suggestions.

**Regarding reproducibility**, we provide the codes for our model in this repository: https://github.com/PatchTST/PatchTST. This is not the final official repo because we want to maintain anonymity. Once the anonymous review ends we will update it with a final official link.

We hope that our replies and revisions address all reviewers' concerns, and we would appreciate any further comments or suggestions.

---

### Comment · Area_Chair_mBVM · 2022-11-22
**Borderline**

Dear Reviewers,

This paper is currently borderline. I also feel borderline about it. It is super simple: For multivariate time series, just take channels independently, split them into patches, run through transformer to predict the entire patch if I understand correctly.

The main positive is that people have tried applying transformer to these problems, but not getting good results, and this paper found the way to do it. The model is not even taking advantage of cross channel information.
The disadvantage as everyone points out is that it is not that novel. It's just 1d patches fed to a transformer for the next patch prediction (or masking).
The paper is also not comparing to S4: https://arxiv.org/pdf/2111.00396.pdf

Any thoughts? In particular given author's responses?

---

> ### Author Response · Authors · 2022-11-22
> **Comparison between PatchTST and S4**
>
> We would like to thank the AC for your comment and suggestion to compare with S4 model. Based on the forecasting results provided in the S4 paper Table 14, we provide the following table to compare the performance of the two models on multivariate setting. It can be seen from the table that the forecasting improvement of PatchTST is significant. The average reduction on MSE is **59.5%** and the average reduction on MAE is **47.8%**.
>
> ||   |  PatchTST  | | S4|  |
> |---|---|---|---|---|---|
> || $T$ |MSE|MAE|MSE|MAE|
> |ETTh1|24| **0.321** | **0.372** | 0.525 | 0.542 |
> ||48| **0.345** | **0.385** | 0.641 | 0.615 |
> ||168| **0.404** | **0.429** | 0.980 | 0.779 |
> ||336|**0.422**|**0.423**|1.407|0.910|
> ||720|**0.447**|**0.468**|1.162|0.842|
> |ETTh2|24| **0.169** | **0.266** | 0.871 | 0.736 |
> ||48| **0.218** | **0.300** |1.240 | 0.867 |
> ||168| **0.325** | **0.371** | 2.580 | 1.255 |
> ||336|**0.329**|**0.384**|1.980|1.128|
> ||720|**0.379**|**0.422**|2.650|1.340|
> |ETTm1|24| **0.198** | **0.278** | 0.426 | 0.487 |
> ||48| **0.261** | **0.321** | 0.580 | 0.565 |
> ||96| **0.293** | **0.346** | 0.699 | 0.649 |
> ||228| **0.353** | **0.386** | 0.824 | 0.674 |
> ||672| **0.409** | **0.416** | 0.846 | 0.709 |
> |Weather|24| **0.090** | **0.123** | 0.334| 0.385 |
> ||48| **0.117** | **0.160** | 0.406 | 0.444 |
> ||168| **0.181** | **0.231** | 0.525 | 0.527 |
> ||336|**0.245**|**0.282**|0.531|0.539|
> ||720|**0.314**|**0.334**|0.578|0.578|
> |ECL|48| **0.110** | **0.205** | 0.255 | 0.352 |
> ||168| **0.145** | **0.239** | 0.283 | 0.373 |
> ||336|**0.163**|**0.259**|0.292|0.382|
> ||720|**0.197**|**0.290**|0.289|0.377|
> ||960|**0.218**|**0.307**|0.299|0.387|

---

### Public Comment · ~Yushu_Chen1 · 2023-02-03
**Some question about ablation study (Table 7)**

The paper is impressive since it improved the performance greatly with a simple approach. We think it is meaningful for applications and future studies.

We still need to ask a question. In the ablation study (Table 7), the original TST also outperforms FEDformer by a large margin in the Weather dataset. We infer that TST is a classic Transformer for time series without the techniques proposed in this paper. However, most of the recent papers show that the accuracy of FEDformer is better than Transformer in time series prediction. Are there some important techniques in TST to promote the performance in additional to a vanilla Transformer?

---

> ### Author Response · Authors · 2023-02-03
> **Reply**
>
> Thanks for the question. We also notice this when we do the experiments. Here are some possible reasons in our mind:
>
> 1.Transformer decoder may lead to performance degrading. This interesting point has been argued in another recent paper (https://arxiv.org/pdf/2212.02789.pdf). TST uses Transformer encoders and a simple head rather than Transformer decoders.
>
> 2.Look-back window. Most of the previous papers are benchmarking their results with look-back window L = 96. In our figure 2, we also show that FEDformer outperforms Transformer when L = 96 (here the Transformer is implemented with decoders). In our table 7, the result for FEDformer is chosen from the best one among six different look-back windows to avoid under-estimation.
>
> 3.Implementation difference. The “original” in table 7 is implemented on channel-mixing TST with patch_len = 1, and numerical difference might exist. Also, FEDformer outperforms TST in other datasets in table 10.

---

> > ### Public Comment · ~Yushu_Chen1 · 2023-02-03
> > **Thanks**
> >
> > Thank you for the reply, which helps us to further understand the paper.

---

### Author Response · Authors · 2023-02-03
**Official Code Link**

We would like to share the official implementation repo to replace the anonymous one since the blind review ends: https://github.com/yuqinie98/PatchTST.

---

### Decision · Program_Chairs · 2023-01-20

**Decision:**

Accept: poster

**Justification For Why Not Higher Score:**

Not large amount of novelty and not a large scale problem.

**Justification For Why Not Lower Score:**

Simple method of applying transformer where previous such applications achieved poor results.

**Metareview: Summary, Strengths And Weaknesses:**

The paper applies transformer to long term forecasting problems of multi-dimensional time series. The method is very simple: Take channels independently, break them into patches and predict the patches into the future using the transformer. The main advantage of this paper is that previous papers have applied transformers to this problem but it resulted in a very weak performance, being beaten by a simple linear methods. This paper found a way to apply the transformer successfully, beating the previous methods.

While there isn't a large amount of novelty, the simplicity is actually a plus, given that the previous approaches using transformers failed and that the method achieves state of the art. To strengthen the paper, the authors should design a transformer architecture that takes in all the channels (not considering them independently) while preserving the advantage of the patch setup they are using and thus improving the performance further (beyond the independent method).

**Note From Pc:**

if the above contains the word "oral" or "spotlight" please see: "oral" presentation means -> notable-top-5% and "spotlight" means -> notable-top-25%. As stated in our emails, we are disassociating presentation type from AC recommendations

**Summary Of Ac-Reviewer Meeting:**

I was only able to meet with one reviewer, given difficulty of scheduling, but my meta-review reflects this.